# Mitigating Source Bias for Fairer Weak Supervision

**Changho Shin, Sonia Cromp, Dyah Adila, Frederic Sala**

Department of Computer Sciences
University of Wisconsin-Madison
{cshin23, cromp, adila, fsala}@wisc.edu

## Abstract

Weak supervision enables efficient development of training sets by reducing the need for ground truth labels. However, the techniques that make weak supervision attractive—such as integrating any source of signal to estimate unknown labels—also entail the danger that the produced pseudolabels are highly biased. Surprisingly, given everyday use and the potential for increased bias, weak supervision has not been studied from the point of view of fairness. We begin such a study, starting with the observation that even when a fair model can be built from a dataset with access to ground-truth labels, the corresponding dataset labeled via weak supervision can be arbitrarily unfair. To address this, we propose and empirically validate a model for source unfairness in weak supervision, then introduce a simple counterfactual fairness-based technique that can mitigate these biases. Theoretically, we show that it is possible for our approach to simultaneously improve both accuracy and fairness—in contrast to standard fairness approaches that suffer from tradeoffs. Empirically, we show that our technique improves accuracy on weak supervision baselines by as much as 32% while reducing demographic parity gap by 82.5%. A simple extension of our method aimed at maximizing performance produces state-of-the-art performance in five out of ten datasets in the WRENCH benchmark.

## 1 Introduction

Weak supervision (WS) is a powerful set of techniques aimed at overcoming the labeled data bottleneck [RSW+16, FCS+20, SLV+22]. Instead of manually annotating points, users assemble noisy label estimates obtained from multiple sources, model them by learning source accuracies, and combine them into a high-quality pseudolabel to be used for downstream training. All of this is done without any ground truth labels. Simple, flexible, yet powerful, weak supervision is now a standard component in machine learning workflows in industry, academia, and beyond [BRL+19]. Most excitingly, WS has been used to build models deployed to billions of devices.

Real-life deployment of models, however, raises crucial questions of fairness and bias. Such questions are tackled in the burgeoning field of fair machine learning [DHP+12, HPS16]. However, weak supervision **has not been studied from this point of view**. This is not a minor oversight. The properties that make weak supervision effective (i.e., omnivorously ingesting any source of signal for labels) are precisely those that make it likely to suffer from harmful biases. This motivates the need to understand and mitigate the potentially disparate outcomes that result from using weak supervision.

The starting point for this work is a simple result. Even when perfectly fair classifiers are possible when trained on ground-truth labels, weak supervision-based techniques can nevertheless produce arbitrarily unfair outcomes. Because of this, simply applying existing techniques for producing fair outcomes to the datasets produced via WS is insufficient—delivering highly suboptimal datasets. Instead, a new approach, specific to weak supervision, must be developed. We introduce a simple technique for improving the fairness properties of weak supervision-based models. Intuitively, a major cause of bias

37th Conference on Neural Information Processing Systems (NeurIPS 2023).

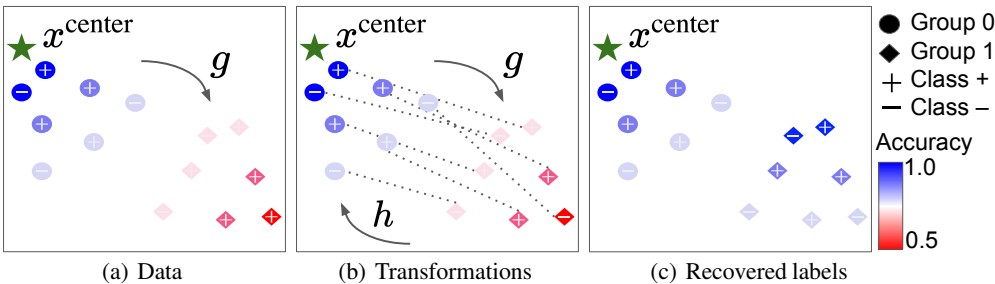

|  | |
|---|---|
| (a) Data | (b) Transformations | (c) Recovered labels |

Figure 1: Intuitive illustration for our setting and approach. (a): circles and diamonds are data points from group 0 and 1, respectively. The accuracy of labeling function is colored-coded, with blue being perfect (1.0) and red random (0.5). Note that accuracy degrades as data points get farther from center $x^{center}$ (star). (b) We can think of group 1 as having been moved far from the center by a transformation $g$, producing lower-quality estimates and violating fairness downstream. (c) Our technique uses counterfactual fairness to undo this transformation, obtaining higher quality estimates. and improved fairness.

in WS is that particular sources are targeted at certain groups, and so produce far more accurate label estimates for these groups—and far more noise for others. We counterfactually ask what outgroup points would most be like if they were part of the 'privileged' group (with respect to each source), enabling us to borrow from the more powerful signal in the sources applied to this group. Thus, the problem is reduced to finding a transformation between groups that satisfies this counterfactual. Most excitingly, while in standard fairness approaches there is a typical tradeoff between fairness and accuracy, with our approach, both the fairness and performance of WS-based techniques can be (sometimes dramatically) improved.

Theoretically, in certain settings, we provide finite-sample rates to recover the counterfactual transformation. Empirically, we propose several ways to craft an efficiently-computed transformation building on optimal transport and some simple variations. We validate our claims on a diverse set of experiments. These include standard real-world fairness datasets, where we observe that our method can improve both fairness and accuracy by as much as 82.5% and 32.5%, respectively, versus weak supervision baselines. Our method can also be combined with other fair ML methods developed for fully supervised settings, further improving fairness. Finally, our approach has implications for WS beyond bias: we combined it with slice discovery techniques [EVS+22] to improve latent underperforming groups. This enabled us to **improve on state-of-the-art on the weak supervision benchmark** WRENCH [ZYL+21].

The contributions of this work include,

- The first study of fairness in weak supervision,
- A new empirically-validated model for weak supervision that captures labeling function bias,
- A simple counterfactual fairness-based correction to mitigate such bias, compatible with any existing weak supervision pipeline, as well as with downstream fairness techniques,
- Theoretical results showing that (1) even with a fair dataset, a weakly-supervised counterpart can be arbitrarily biased and (2) a finite-sample recovery result for the proposed algorithm,
- Experiments validating our claims, including on weakly-supervised forms of popular fairness evaluation datasets, showing gains in fairness metrics—and often simultaneously improvements in accuracy.

## 2 Background and Related Work

We present some high-level background on weak supervision and fairness in machine learning. Afterward, we provide setup and present the problem statement.

**Weak Supervision** Weak supervision frameworks build labeled training sets *with no access to ground truth labels*. Instead, they exploit multiple sources that provide noisy estimates of the label. These sources include heuristic rules, knowledge base lookups, pretrained models, and more [KOS11, MBSJ09, GM14, DZS+17, RBE+18]. Because these sources may have different—and unknown—accuracies and dependencies, their outputs must be modeled in order to produce a combination that can be used as a high-quality pseudolabel.

Concretely, there is a dataset $\{(x_1,y_1),...,(x_n,y_n)\}$ with unobserved true label $y_i \in \{-1,+1\}$. We can access the outputs of $m$ sources (labeling functions) $\lambda^1,\lambda^2,...,\lambda^m : \mathcal{X} \rightarrow \{-1,+1\}$. These outputs are modeled via a generative model called the *label model*, $p_\theta(\lambda^1,...,\lambda^m,y)$. The goal is to estimate the parameters $\theta$ of this model, without accessing the latent $y$, and to produce a pseudolabel estimate $p_{\hat{\theta}}(y|\lambda^1,...,\lambda^m)$. For more background, see [ZHY+22].

**Machine Learning and Fairness** Fairness in machine learning is a large and active field that seeks to understand and mitigate biases. We briefly introduce high-level notions that will be useful in the weak supervision setting, such as the notion of fairness metrics. Two popular choices are demographic parity [DHP+12] and equal opportunity [HPS16]. Demographic parity is based on the notion that individuals of different groups should have equal treatment, i.e., if $A$ is the group attribute, $P(\hat{Y}=1|A=1) = P(\hat{Y}=1|A=0)$. The equal opportunity principle requires that predictive error should be equal across groups, i.e., $P(\hat{Y}=1|Y=1,A=1) = P(\hat{Y}=1|Y=1,A=0)$. A large number of works study, measure, and seek to improve fairness in different machine learning settings based on these metrics. Typically, the assumption is that the underlying dataset differs within groups in such a way that a trained model will violate, for example, the equal opportunity principle. In contrast, in this work, we focus on additional violations of fairness that are induced by weak supervision pipelines—which can create substantial unfairness even when the true dataset is perfectly fair. In the same spirit, [WH22] considers fairness in positive-and-unlabeled (PU) settings, where true labels are available, but only for one class, while other points are unlabeled. Another related line of research is fairness under noisy labels [WLL21, KL22, WZN+23, ZZL+23]. These works consider the noise rate of labels in fair learning, enhancing the robustness of fair learning methods. *A crucial difference between such works and ours: in weak supervision, we have multiple sources of noisy labels—and we can exploit these to directly improve dataset fairness.*

**Counterfactual Fairness** Most closely related to the notion of fairness we use in this work is *counterfactual fairness*. [KLRS17] introduced such a counterfactual fairness notion, which implies that changing the sensitive attribute $A$, while keeping other variables causally not dependent on $A$, should not affect the outcome. While this notion presumes the causal structure behind the ML task, it is related to our work in the sense that our proposed method tries to remove the causal effect by $A$ with particular transformations. A more recent line of work has proposed bypassing the need for causal structures and directly tackling counterfactual fairness through optimal transport [GDBFL19, BYF20, SRT+20, SMBN21, BDB22]. The idea is to detect or mitigate unfairness by mapping one group to another group via such techniques. In this paper, we build on these tools to help improve fairness while avoiding the accuracy-fairness tradeoff common to most settings.

## 3   Mitigating Labeling Function-Induced Unfairness

We are ready to explain our approach to mitigating unfairness in weak supervision sources. First, we provide a flexible model that captures such behavior, along with empirical evidence supporting it. Next, we propose a simple solution to correct unfair source behavior via optimal transport.

**Modeling Group Bias in Weak Supervision** Weak supervision models the accuracies and correlations in labeling functions. The standard model, used in [RHD+19a, FCS+20] and others is $P(\lambda^1,...,\lambda^m,y) = \frac{1}{Z}\exp(\theta_y y + \sum_{j=1}^m \theta_j \lambda^j y)$, with $\theta_j \geq 0$. We leave out the correlations for simplicity; all of our discussion below holds when considering correlations as well. Here, $Z$ is the normalizing partition function. The $\theta$ are *canonical parameters* for the model. $\theta_y$ sets the class balance. The $\theta_i$'s capture how accurate labeling function (LF) $i$ is: if $\theta_i = 0$, the LF produces random guesses. If $\theta_i$ is relatively large, the LF is highly accurate. A weakness of this model is that it *ignores the feature vector* $x$. It implies that LFs are uniformly accurate over the feature space—a highly unrealistic assumption. A more general model was presented in [CFA+22], where there is a model for each feature vector $x$, i.e.,

$$P_x(\lambda^1,...,\lambda^m,y) = \frac{1}{Z}\exp(\theta_y y + \sum_{j=1}^m \theta_{j,x} \lambda^j(x)y). \tag{1}$$

However, as we see only one sample for each $x$, it is impossible to recover the parameters $\theta_x$. Instead, the authors assume a notion of *smoothness*. This means that the $\theta_{j,x}$'s do not vary in small neighborhoods, so that the feature space can be partitioned and a single model learned per part. Thus

*model* (1) *from [CFA$^+$22] is more general, but still requires a strong smoothness assumption.* It also does not encode any notion of bias. Instead, we propose a model that encodes both smoothness and bias.

Concretely, assume that the data are drawn from some distribution on $\mathcal{Z} \times \mathcal{Y}$, where $\mathcal{Z}$ is a latent space. We do not observe samples from $\mathcal{Z}$. Instead, there are $l$ transformation functions $g_1, ..., g_l$, where $g_k : \mathcal{Z} \to \mathcal{X}$. For each point $z_i$, there is an assigned group $k$ and we observe $x_i = g_k(z_i)$. Then, our model is the following:

$$P(\lambda^1(z),...,\lambda^m(z),y) = \frac{1}{Z} \exp \left( \theta_y y + \sum_{j=1}^{m} \frac{\theta_j}{1 + d(x^{\text{center}_j}, g_k(z))} \lambda^j(g_k(z)) y \right). \qquad (2)$$

We explain this model as follows. We can think of it as a particular version of (1). However, instead of arbitrary $\theta_{j,x}$ parameters for each $x$, we explicitly model these parameters as two components: a feature-independent accuracy parameter $\theta_j$ and a term that modulates the accuracy based on the distance between feature vector $x$ and some fixed center $x^{\text{center}_j}$. The center represents, for each LF, a *most accurate point*, where accuracy is maximized at a level set by $\theta_j$. As the feature vector $x = g_k(z)$ moves away from this center, the denominator $1 + d(x^{\text{center}_j}, g_k(z))$ increases, and the LF votes increasingly poorly. This is an explicit form of smoothness that we validate empirically below.

For simplicity, we assume that there are two groups, indexed by $0,1$, that $\mathcal{X} = \mathcal{Z}$, and that $g_0(z) = z$. In other words, the transformation for group 0 is the identity, while this may not be the case for group 1. Simple extensions of our approach can handle cases where none of these assumptions are met.

**Labeling Function Bias** The model (2) explains how and when labeling functions might be biased. Suppose that $g_k$ takes points $z$ far from $x^{\text{center}_j}$. Then, the denominator term in (2) grows—and so the penalty for $\lambda(x)$ to disagree with $y$ is reduced, making the labeling function less accurate. This is common in practice. For example, consider a situation where a bank uses features that include credit scores for loan review. Suppose that the group variable is the applicant's nationality. Immigrants typically have a shorter period to build credit; this is reflected in a transformed distribution $g_1(z)$. A labeling function using a credit score threshold may be accurate for non-immigrants, but may end up being highly inaccurate when applied to immigrants.

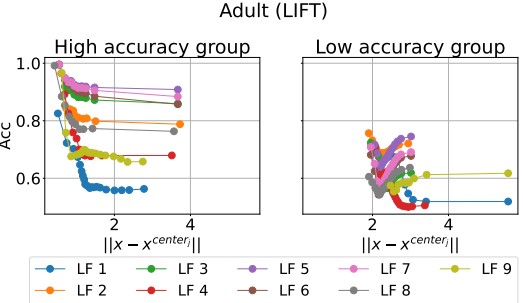

Figure 2: Average accuracy (y-axis) depending on the distance to the center point (x-axis). The center is obtained by evaluating the accuracy of their neighborhood data points.

We validate this notion empirically. We used the Adult dataset [K$^+$96], commonly used for fairness studies, with a set of custom-built labeling functions. In Figure 2, we track the accuracies of these LFs as a function of distance from an empirically-discovered center $x^{\text{center}_j}$. On the left is the high-accuracy group in each labeling function; as expected in our model, as the distance from the center $\|x - x^{center_j}\|$ is increased, the accuracy decreases. On the right-hand side, we see the lower-accuracy group, whose labeling functions are voting $x_i = g_1(z_i)$. This transformation has sent these points further away from the center (note the larger distances). As a result, the overall accuracies have also decreased. Note, for example, how LF 5, in purple, varies between 0.9 and 1.0 accuracy in one group and is much worse—between 0.6 and 0.7—in the other.

## 3.1 Correcting Unfair LFs

Given the model (2), how can we reduce the bias induced by the $g_k$ functions? A simple idea is to *reverse* the effect of the $g_k$'s. If we could invert these functions, violations of fairness would be mitigated, since the accuracies of labeling functions would be uniformized over the groups.

Concretely, suppose that $g_k$ is invertible and that $h_k$ is this inverse. If we knew $h_k$, then we could ask the labeling functions to vote on $h_k(x) = h_k(g_k(x)) = z$, rather than on $x = g_k(z)$, and we could do so for any group, yielding equal-accuracy estimates for all groups. The technical challenge is how to estimate the inverses of the $g_k$'s, without any parametric form for these functions. To do so, we deploy optimal transport (OT) [PC$^+$19]. OT transports a probability distribution to another probability

---
**Algorithm 1:** SOURCE BIAS MITIGATION (SBM)
---

1: **Parameters:** Features $X_0$, $X_1$ and LF outputs $\Lambda_0 = [\lambda_0^1, ..., \lambda_0^m]$, $\Lambda_1 = [\lambda_1^1, ..., \lambda_1^m]$ for groups 0, 1, transport threshold $\varepsilon$
2: **Returns:** Modified weak labels $\Lambda = [\lambda^1, ..., \lambda^m]$
3: Estimate accuracy of $\lambda^j$ in each group, $\hat{a}_0^j = \hat{\mathbb{E}}[\lambda_j Y | A = 0]$, $\hat{a}_1^j = \hat{\mathbb{E}}[\lambda_j Y | A = 1]$ from $\Lambda_0, \Lambda_1$ with Algorithm 2
4: **for** $j \in \{1, 2, ..., m\}$ **do**
5:     **if** $\hat{a}_1^j \geq \hat{a}_0^j + \varepsilon$ **then** update $\lambda_0^j$ by transporting $X_0$ to $X_1$ (Algorithm 3)
6:     **else if** $\hat{a}_0^j \geq \hat{a}_1^j + \varepsilon$ **then** update $\lambda_1^j$ by transporting $X_1$ to $X_0$ (Algorithm 3)
7: **end for**
8: **return** $\Lambda = [\lambda^1, ..., \lambda^m]$

---

distribution by finding a minimal cost coupling. We use OT to recover the reverse map $h_k : \mathcal{X} \to \mathcal{Z}$ by $\hat{h}_k = \arg\inf_{T_\sharp \nu = \omega} \{ \int_{x \in \mathcal{X}} c(x, T(x)) d\nu(x) \}$, where $c$ is a cost functon, $\nu$ is a probability measure in $\mathcal{X}$ and $\omega$ is a probability measure in $\mathcal{Z}$.

Our proposed approach, building on the use of OT, is called *source bias mitigation* (SBM). It seeks to reverse the group transformation $g_k$ via OT. The core routine is described in Algorithm 1. The first step of the algorithm is to estimate the accuracies of each group so that we can identify which group is privileged, i.e., which of the transformations $g_0, g_1$ is the identity map. To do this, we use Algorithm 2 [FCS$^+$20] by applying it to each group separately.

After identifying the high-accuracy group, we transport data points from the low-accuracy group to it. Since not every transported point perfectly matches an existing high-accuracy group point, we find a nearest neighbor and borrow its label. We do this only when there is a sufficient inter-group accuracy gap, since the error in transport might otherwise offset the benefit. In practice, if the transformation is sufficiently weak, it is possible to skip optimal transport and simply use nearest neighbors. Doing this turned out to be effective in some experiments (Section 5.1). Finally, after running SBM, modified weak labels are used in a standard weak supervision pipeline, which is described in Appendix C.

## 4 Theoretical Results

We provide two types of theoretical results. First, we show that labeling function bias can be arbitrarily bad—resulting in substantial unfairness—regardless of whether the underlying dataset is fair. Next, we show that in certain settings, we can consistently recover the fair labeling function performance when using Algorithm 1, and provide a finite-sample error guarantee. Finally, we comment on extensions. All proofs are located in Appendix D.

**Setting and Assumptions** We assume that the distributions $P_0(x)$ and $P_1(x')$ are subgaussian with means $\mu_0$ and $\mu_1$ and positive-definite covariance matrices $\Sigma_0$ and $\Sigma_1$, respectively. Note that by assumption, $P_0(x) = P(z)$ and $P_1(x')$ is the pushforward of $P_0(x)$ under $g_1$. Let $\mathbf{r}(\Sigma)$ denote the effective rank of $\Sigma$ [Ver18]. We observe $n_0$ and $n_1$ i.i.d. samples from groups 0 and 1, respectively. We use Euclidean distance as the distance $d(x, y) = \|x - y\|$ in model (2). For the unobserved ground truth labels, $y_i$ is drawn from some distribution $P(y|z)$. Finally, the labeling functions voting on our points are drawn via the model (2).

### 4.1 Labeling Functions can be Arbitrarily Unfair

We show that, as a result of the transformation $g_1$, the predictions of labeling functions can be arbitrarily unfair even if the dataset is fair. The idea is simple: the average group 0 accuracy, $\mathbb{E}_{z \in \mathcal{Z}}[P(\lambda(I(z)) = y)]$, is independent of $g_1$, so it suffices to show that $\mathbb{E}_{x' \in g_1(\mathcal{Z})}[P(\lambda(x') = y)]$ can deteriorate when $g_1$ moves data points far from the center $x^{\text{center}_0}$. As such, we consider the change in $\mathbb{E}_{x' \in g_1(\mathcal{Z})}[P(\lambda(x') = y)]$ as the group 1 points are transformed increasingly far from $x^{center_0}$ in expectation.

**Theorem 4.1.** *Let $g_1^{(k)}$ be an arbitrary sequence of functions such that $\lim_{k \to \infty} \mathbb{E}_{x' \in g_1^{(k)}(\mathcal{Z})}[\|x' - x^{center_0}\|] \to \infty$. Suppose our assumptions above are met; in particular, that the label $y$ is independent of the observed features $x = I(z)$ or $x' = g_1^{(k)}(z), \forall k$, conditioned*

Table 1: Tabular dataset results

| | Adult | | | | Bank Marketing | | | |
|---|---|---|---|---|---|---|---|---|
| | Acc ($\uparrow$) | F1 ($\uparrow$) | $\Delta_{DP}$ ($\downarrow$) | $\Delta_{EO}$ ($\downarrow$) | Acc ($\uparrow$) | F1 ($\uparrow$) | $\Delta_{DP}$ ($\downarrow$) | $\Delta_{EO}$ ($\downarrow$) |
| FS | 0.824 | 0.564 | 0.216 | 0.331 | 0.912 | 0.518 | 0.128 | 0.117 |
| WS (Baseline) | 0.717 | 0.587 | 0.475 | 0.325 | 0.674 | 0.258 | 0.543 | 0.450 |
| SBM (w/o OT) | 0.720 | **0.592** | 0.439 | 0.273 | 0.876 | **0.550** | 0.106 | **0.064** |
| SBM (OT-L) | 0.560 | 0.472 | 0.893 | 0.980 | **0.892** | 0.304 | 0.095 | 0.124 |
| SBM (OT-S) | 0.723 | 0.590 | 0.429 | 0.261 | 0.847 | 0.515 | 0.122 | 0.080 |
| SBM (w/o OT) + LIFT | 0.704 | 0.366 | 0.032 | 0.192 | 0.698 | 0.255 | **0.088** | 0.137 |
| SBM (OT-L) + LIFT | 0.700 | 0.520 | 0.015 | **0.138** | **0.892** | 0.305 | 0.104 | 0.121 |
| SBM (OT-S) + LIFT | **0.782** | 0.448 | **0.000** | 0.178 | 0.698 | 0.080 | 0.109 | 0.072 |

*on the latent features $z$. Then,*

$$\lim_{k \to \infty} \mathbb{E}_{x' \in g_1^{(k)}(\mathcal{Z})}[P(\lambda(x') = y)] = \frac{1}{2},$$

*which corresponds to random guessing.*

It is easy to construct such a sequence of functions $g_1^{(k)}$, for instance, by letting $g_1^{(k)}(z) = z + ku$, where $u$ is a $d$-dimensional vector of ones. When the distribution of group 1 points lies far from $x^{\text{center}_0}$ while the distribution of group 0 points lies near to $x^{\text{center}_0}$, the accuracy parity of $\lambda$ suffers. With adequately large expected $d(x^{\text{center}_0}, g_1^{(k)}(z))$, the performance of $\lambda$ on group 1 points approaches random guessing.

### 4.2 Finite-Sample Bound for Mitigating Unfairness

Next, we provide a result bounding the difference in LF accuracy between group 0 points, $\mathbb{E}_{x \in \mathcal{Z}}[P(\lambda(x) = y)]$, and group 1 points transformed using our method, $\mathbb{E}_{x' \in \mathcal{X}}[P(\lambda(\hat{h}(x')) = y)]$. A tighter bound on this difference corresponds to better accuracy intra-group parity.

**Theorem 4.2.** *Set $\tau$ to be $\max\big(\mathbf{r}(\Sigma_0)/n_0, \mathbf{r}(\Sigma_1)/n_1, t/\min(n_0, n_1), t^2/\max(n_0, n_1)^2\big)$, and let $C$ be a constant. Under the assumptions described above, when using Algorithm 1, for any $t > 0$, we have that with probability $1 - e^{-t} - 1/n_1$,*

$$|\mathbb{E}_{x \in \mathcal{Z}}[P(\lambda(x) = y)] - \mathbb{E}_{x' \in \mathcal{X}}[P(\lambda(\hat{h}(x')) = y)]| \le 4\theta_0 C \sqrt{\tau \mathbf{r}(\Sigma_1)},$$

Next we interpret Theorem 4.2. LF accuracy recovery scales with $\max\big(1/\sqrt{n_0}, 1/\sqrt{n_1}\big)$. This does not present any additional difficulties compared to vanilla weak supervision—it is the same rate we need to learn LF accuracies. In other words, there is no sample complexity penalty for using our approach. Furthermore, LF accuracy recovery scales inversely to $\max\big(\sqrt{\mathbf{r}(\Sigma_0)\mathbf{r}(\Sigma_1)}, \mathbf{r}(\Sigma_1)\big)$. That is, when the distributions $P_0(x)$ or $P_1(x')$ have greater spread, it is more difficult to restore fair behavior.

Finally, we briefly comment on extensions. It is not hard to extend these results to a setting with less strict assumptions. For example, we can take $P$ to be a mixture of Gaussians. In this case, it is possible to combine algorithms for learning mixtures [CGT18] with the approach we presented.

## 5 Experiments

The primary objective of our experiments is to validate that SBM improves fairness while often enhancing model performance as well. In real data experiments, we confirm that our methods work well with real-world fairness datasets (Section 5.1). In the synthetic experiments, we validate our theory claims in a fully controllable setting—showing that our method can achieve perfect fairness and performance recovery (Section 5.2). In addition, we show that our method **is compatible with other fair ML techniques** developed for fully supervised learning (Section 5.3). Finally, we demonstrate that our

Table 2: NLP dataset results            Table 3: Vision dataset results

| Dataset | Methods | Acc | F1 | $\Delta_{DP}$ | $\Delta_{EO}$ | Dataset | Methods | Acc | F1 | $\Delta_{DP}$ | $\Delta_{EO}$ |
|---|---|---|---|---|---|---|---|---|---|---|---|
| **Civil** | FS | 0.893 | 0.251 | 0.083 | 0.091 | **CelebA** | FS | 0.897 | 0.913 | 0.307 | 0.125 |
| | WS (Baseline) | 0.854 | **0.223** | 0.560 | 0.546 | | WS (Baseline) | 0.866 | 0.879 | 0.308 | 0.193 |
| | SBM (w/o OT) | 0.879 | 0.068 | 0.048 | 0.047 | | SBM (w/o OT) | 0.870 | 0.883 | 0.309 | 0.192 |
| | SBM (OT-L) | 0.880 | 0.070 | 0.042 | 0.039 | | SBM (OT-L) | 0.870 | 0.883 | **0.306** | 0.185 |
| | SBM (OT-S) | **0.882** | 0.047 | **0.028** | **0.026** | | SBM (OT-S) | **0.872** | **0.885** | **0.306** | **0.184** |
| **Hate** | FS | 0.698 | 0.755 | 0.238 | 0.121 | **UTKF** | FS | 0.810 | 0.801 | 0.133 | 0.056 |
| | WS (Baseline) | 0.584 | 0.590 | 0.170 | 0.133 | | WS (Baseline) | 0.791 | 0.791 | 0.172 | 0.073 |
| | SBM (w/o OT) | 0.592 | 0.637 | 0.159 | 0.138 | | SBM (w/o OT) | 0.797 | 0.790 | 0.164 | 0.077 |
| | SBM (OT-L) | **0.670** | 0.606 | 0.120 | 0.101 | | SBM (OT-L) | 0.800 | 0.793 | 0.135 | 0.043 |
| | SBM (OT-S) | 0.612 | **0.687** | **0.072** | **0.037** | | SBM (OT-S) | **0.804** | **0.798** | **0.130** | **0.041** |

method can improve weak supervision performance beyond fairness by applying techniques to discover underperforming data slices (Section 5.4). This enables us to outperform state-of-the-art on a popular weak supervision benchmark [ZYL+21]. Our code is available at https://github.com/SprocketLab/fair-ws.

## 5.1 Real data experiments

**Claims Investigated**    In real data settings, we hypothesize that our methods can reduce the bias of LFs, leading to better fairness and improved performance of the weak supervision end model.

**Setup and Procedure**    We used 6 datasets in three different domains: tabular (Adult and Bank Marketing), NLP (CivilComments and HateXplain), and vision (CelebA and UTKFace). Their task and group variables are summarized in Appendix E, Table 7. LFs are either heuristics or pretrained models. More details are included in Appendix E.3.

For the weak supervision pipeline, we followed a standard procedure. First, we generate weak labels from labeling functions in the training set. Secondly, we train the label model on weak labels. In this experiment, we used Snorkel [BRL+19] as the label model in weak supervision settings. Afterwards, we generate pseudolabels from the label model, train the end model on these, and evaluate it on the test set. We used logistic regression as the end model. The only difference between our method and the original weak supervision pipeline is a procedure to fix weak labels from each labeling function. As a sanity check, a fully supervised learning result (FS), which is the model performance trained on the true labels, is also provided. Crucially, however, *in weak supervision, we do not have such labels*, and therefore fully supervised learning is simply an upper bound to performance—and not a baseline.

We ran three variants of our method. *SBM (w/o OT)* is a 1-nearest neighbor mapping to another group without any transformation. *SBM (OT-L)* is a 1-nearest neighbor mapping with a linear map learned via optimal transport. *SBM (OT-S)* is a 1-nearest neighbor mapping with a barycentric mapping learned via the Sinkhorn algorithm. To see if our method can improve both fairness and performance, we measured the demographic parity gap ($\Delta_{DP}$) and the equal opportunity gap ($\Delta_{EO}$) as fairness metrics, and computed accuracy and F1 score as performance metrics as well.

**Results**    The tabular dataset result is reported in Table 1. As expected, our method improves accuracy while reducing demographic parity gap and equal opportunity gap. However, we observed *SBM (OT-L)* critically fails at Adult dataset, contrary to what we anticipated. We suspected this originates in one-hot coded features, which might distort computing distances in the nearest neighbor search. To work around one-hot coded values in nearest neighbor search, we deployed LIFT [DZZ+22], which encodes the input as natural language (e.g. "She/he is <race attribute>. She/he works for <working hour attribute> per week ...") and embeds them with language models (LMs). We provide heuristic rules to convert feature columns into languages in Appendix E.2, and we used BERT as the language model. The result is given in Table 1 under the dashed lines. While it sacrifices a small amount of accuracy, it substantially reduces the unfairness as expected.

The results for NLP datasets are provided in Table 2. In the CivilComments and HateXplain datasets, we observed our methods mitigate bias consistently, as we hoped. While our methods improve performance

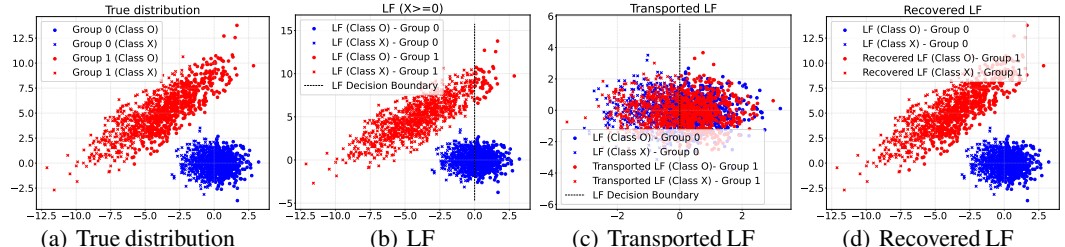

| (a) True distribution | (b) LF | (c) Transported LF | (d) Recovered LF |

Figure 3: Synthetic datasets. In (a), seemingly different data distributions from the two groups actually have perfect achievable fairness. However, the labeling function in (b) only works well in group 0, which leads to unfairness. Via OT (c), the input distribution can be matched and the LF applied to similar groups—original and recovered. As a result, LFs on group 1 works as well as on 0 (d).

as well in the HateXplain dataset, enhancing other metrics in CivilComments results in drops in the F1 score. We believe that a highly unbalanced class setting ($P(Y=1)\approx 0.1$) is the cause of this result.

The results for vision datasets are given in Table 3. Though not as dramatic as other datasets since here the LFs are pretrained models, none of which are heavily biased, our methods can still improve accuracy and fairness. In particular, our approach shows clear improvement over the baseline, which yields performance closer to the fully supervised learning setting while offering less bias.

## 5.2 Synthetic experiments

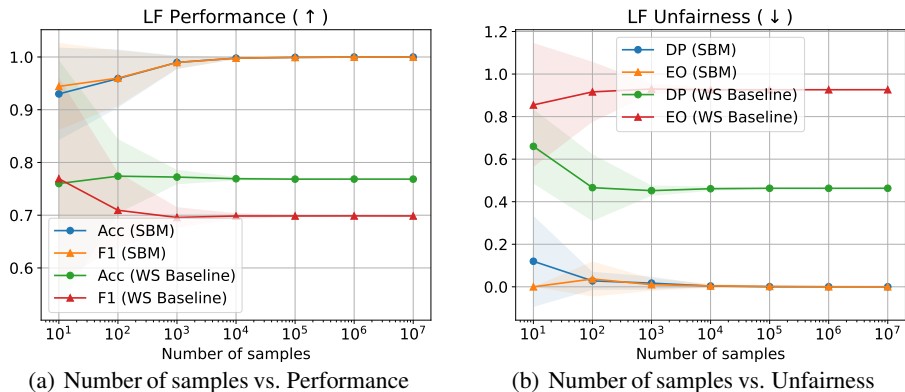

| (a) Number of samples vs. Performance | (b) Number of samples vs. Unfairness |

Figure 4: Synthetic experiment on number of samples vs. performance and fairness. Confidence intervals are obtained by $\pm 1.96\times$ standard deviation of 10 repetition with different seeds.

**Claim Investigated** We hypothesized that our method can recover both fairness and accuracy (as a function of the number of samples available) by transporting the distribution of one group to another group when our theoretical assumptions are almost exactly satisfied. To show this, we generate unfair synthetic data and LFs and see if our method can remedy LF fairness and improve LF performance.

**Setup and Procedure** We generated a synthetic dataset that has perfect fairness as follows. First, $n$ input features in $\mathbb{R}^2$ are sampled as $X_0\sim\mathcal{N}(\mathbf{0},I)$ for group 0, and labels $Y_0$ are set by $Y_0=\mathbb{1}(X_0[0]\geq 0.5)$, i.e. 1 if the first dimension is positive or equal. Afterwards, $n$ input features in $\mathbb{R}^2$ are sampled as $\tilde{X}_1\sim\mathcal{N}(0,I)$ for group 1, and the labels are also set by $Y_1=\mathbb{1}(\tilde{X}_1[0]\geq 0.5)$. Then, a linear transformation is applied to the input distribution: $X_1=\Sigma\tilde{X}_1+\mu$ where $\mu=\begin{bmatrix}-4\\5\end{bmatrix}$, $\Sigma=\begin{bmatrix}2 & 1\\1 & 2\end{bmatrix}$, which is the distribution of group 1. Clearly, we can see that $X_1=\Sigma X_0+\mu\sim\mathcal{N}(\mu,\Sigma)$. Here we applied the same labeling function $\lambda(x)=\mathbb{1}(x[0]\geq 0)$, which is the same as the true label distribution in group 0.

Table 4: Compatibility with other fair ML methods (HateXplain dataset)

| | Acc | F1 | $\Delta_{DP}$ | $\Delta_{EO}$ |
|---|---|---|---|---|
| FS | 0.698 | 0.755 | 0.238 | 0.121 |
| WS (Baseline) | 0.584 | 0.590 | 0.171 | 0.133 |
| SBM (OT-S) | 0.612 | 0.687 | 0.072 | 0.037 |
| WS (Baseline) + OTh-DP | 0.539 | 0.515 | 0.005 | 0.047 |
| SBM (OT-S) + OTh-DP | 0.607 | 0.694 | 0.002 | 0.031 |

Table 5: Slice discovery with SBM results in WRENCH. Evaluation metric is accuracy for iMDb, F1 for the rest.

| Methods | Basketball | Census | iMDb | Mushroom | Tennis |
|---|---|---|---|---|---|
| FS | 0.855 | 0.634 | 0.780 | 0.982 | 0.858 |
| WS (HyperLM) | 0.259 | 0.551 | 0.753 | 0.866 | 0.812 |
| SBM (w/o OT) | 0.261 | 0.568 | 0.751 | 0.790 | 0.819 |
| SBM (OT-L) | 0.242 | 0.547 | 0.756 | 0.903 | 0.575 |
| SBM (OT-S) | 0.260 | 0.552 | 0.756 | 0.935 | 0.663 |

We apply our method (SBM OT-L) since our data model fits its basic assumption. Again, we evaluated the results by measuring accuracy, F1 score, $\Delta_{DP}$, and $\Delta_{EO}$. The setup and procedure are illustrated in Figure 3. We varied the number of samples $n$ from $10^2$ to $10^7$

**Results** The result is reported in Figure 4. As we expected, we saw the accuracy and F1 score are consistently improved as the linear Monge map is recovered when the number of samples $n$ increases. Most importantly, we observed that perfect fairness is achieved after only a small number of samples ($10^2$) are obtained.

### 5.3 Compatibility with other fair ML methods

**Claim Investigated** Our method corrects labeling function bias at the individual LF—and not model—level. We expect our methods can work cooperatively, in a constructive way, with other fair ML methods developed for fully supervised learning settings.

**Setup and Procedure** We used the same real datasets, procedures, and metrics as before. We combined the optimal threshold method [HPS16] with WS (baseline) and our approach, SBM (Sinkhorn). We denote the optimal threshold with demographic parity criteria as OTh-DP.

**Results** The results are shown in Table 4. As we expected, we saw the effect of optimal threshold method, which produces an accuracy-fairness (DP) tradeoff. This has the same effect upon our method. Thus, when optimal threshold is applied to both, our method has better performance and fairness aligned with the result without optimal threshold. More experimental results with other real datasets and additional fair ML methods are reported in Appendix E.5.

### 5.4 Beyond fairness: maximizing performance with slice discovery

**Claim Investigated** We postulated that even outside the context of improving fairness, our techniques can be used to boost the performance of weak supervision approaches. In these scenarios, there are no pre-specified groups. Instead, underperforming latent groups (slices) must first be discovered. Our approach then uses transport to improve labeling function performance on these groups.

**Setup and Procedure** We used Basketball, Census, iMDb, Mushroom, and Tennis dataset from the WRENCH benchmark [ZYL+21], which is a well-known weak supervision benchmark but does not include any group information. We generated group annotations by slice discovery [KGZ19, SNS+21, ddWLB22, EVS+22], which is an approach to discover data slices that share a common characteristic. To find groups with a large accuracy gap, we used Domino [EVS+22]. It discovers regions of the embedding space based on the accuracy of model. Since the WS setting does not allow access to true labels, we replaced true labels with pseudolabels obtained from the label model, and model scores with label model probabilities. In order to show we can increase performance *even for state-of-the-art weak supervision*, we used the recently-proposed state-of-the-art Hyper Label Model [WCZC22] as the label model. We used the group information generated by the two discovered slices to apply our methods. We used logistic regression as the end model, and used the same weak supervision pipeline and metrics as in the other experiments, excluding fairness.

**Results** The results can be seen in Table 5. As expected, even without known group divisions, we still observed improvements in accuracy and F1 score. We see the most significant improvements on the Mushroom dataset, where we substantially close the gap to fully-supervised. These gains suggest that it is possible to generically combine our approach with other principled methods for subpopulation discovery to substantially improve weak supervision in general settings.

# 6 Conclusion

Weak supervision has been successful in overcoming manual labeling bottlenecks, but its impact on fairness has not been adequately studied. Our work has found that WS can easily induce additional bias due to unfair LFs. In order to address this issue, we have proposed a novel approach towards mitigating bias in LFs and further improving model performance. We have demonstrated the effectiveness of our approach using both synthetic and real datasets and have shown that it is compatible with traditional fair ML methods. We believe that our proposed technique can make weak supervision safer to apply in important societal settings and so encourages its wider adoption.

**Acknowledgments**

We are grateful for the support of the NSF under CCF2106707 (Program Synthesis for Weak Supervision) and the Wisconsin Alumni Research Foundation (WARF). We thank Nick Roberts, Harit Vishwakarma, Tzu-Heng Huang, Jitian Zhao, and John Cooper for their helpful feedback and discussion.

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

# Appendix

The appendix contains additional details, proofs, and experimental results. The glossary contains a convenient reminder of our terminology (Appendix A). Appendix B provides more related works and discussion about the relationship between our work and related papers. In Appendix C, we describe the details of our algorithm and discuss their implementations. Appendix D provides the proofs of theorems that appeared in Section 4. Finally, we give more details and analysis of the experiments and provide additional experiment results.

## A  Glossary

The glossary is given in Table 6 below.

| Symbol | Definition |
|---|---|
| $\mathcal{X}$ | Feature space |
| $\mathcal{Y}$ | Label metric space |
| $\mathcal{Z}$ | Latent space of feature space |
| $A$ | Group variable, assumed to have one of two values $\{0,1\}$ for simplicity |
| $X_k$ | Inputs from group $k$. |
| $X$ | Inputs from all groups $k \in \{1,...,l\}$ |
| $Y_k$ | True labels from group $k$ |
| $Y$ | True labels from all groups $k \in \{1,...,l\}$ |
| $P$ | Probability distribution in the latent space |
| $P_k$ | Probability distribution in the group k |
| $\lambda_k^j$ | Noisy labels of LF $j$ of group $k$. If it is used with input (e.g. $\lambda_k^j(x)$), it denotes LF $j$ from group $k$ such that its outputs are noisy labels |
| $\lambda^j$ | Noisy labels of LF $j$ of all groups $k \in \{1,...,l\}$. If it is used with input (e.g. $\lambda^j(x)$), it denotes LF $j$ such that its outputs are noisy labels |
| $\Lambda_k$ | Collection of noisy labels from group $k$, $\Lambda_k = [\lambda_k^1,...,\lambda_k^m]$ |
| $\Lambda$ | Collection of noisy labels from all groups $k \in \{1,...,l\}$, $\Lambda = [\lambda^1,...,\lambda^m]$ |
| $g_k$ | $g_k : \mathcal{Z} \to \mathcal{X}$, $k$-th group transformation function |
| $h_k$ | $h_k : \mathcal{X} \to \mathcal{Z}$ the inverse transformation of $g_k$, i.e. $h_k g_k(x) = x$ for $x \in \mathcal{Z}$ |
| $\theta_y$ | Prior parameter for $Y$ in label model |
| $\theta_j$ | Accuracy parameter for $\lambda^j$ in label model |
| $\theta_{j,x}$ | Accuracy parameter for $x$ of $\lambda^j$ in label model [CFA$^+$22] |
| $a_k^j$ | Accuracy of LF $j$ in group $k$, $a_k^j = \mathbb{E}[\lambda_j Y \mid A = k]$ |
| $a^j$ | Accuracy of LF $j$, $a^j = \mathbb{E}[\lambda_j Y]$ |
| $\hat{a}_k^j, \hat{a}^j$ | Estimates of $a_k^j, a^j$ |
| $\mu_k$ | Mean of features in group $k$ |
| $\Sigma_k$ | Covariance of features in group $k$ |
| $I$ | Identity transformation, i.e. $I(x) = x$ |
| $tr(\Sigma)$ | Trace of $\Sigma$ |
| $\lambda_{\max}(\Sigma), \lambda_{\min}(\Sigma)$ | Maximum, minimum values of $\Sigma$ |
| $\mathbf{r}(\Sigma)$ | Effective rank of $\Sigma$, i.e. $\mathbf{r}(\Sigma) = \frac{tr(\Sigma)}{\lambda_{\max}(\Sigma)}$ |

Table 6: Glossary of variables and symbols used in this paper.

## B  Extended related work

**Weak supervision**   In weak supervision, we assume the true label $Y \in \mathcal{Y}$ cannot be accessed, but labeling functions $\lambda^1, \lambda^2,..., \lambda^m \in \mathcal{Y}$ which are noisy versions of true labels, are provided. These weak label sources include code snippets expressing heuristics about $Y$, crowdworkers, external knowledge bases, pretrained models, and others [KOS11, MBSJ09, GM14, DZS$^+$17, RBE$^+$18]. Given $\lambda^1, \lambda^2,···,\lambda^m$, WS often takes a two-step procedure to produce an end model [DS79, RSW$^+$16, FCS$^+$20, RHD$^+$19b, SLV$^+$22, VS22]. The first step, the label model, obtains fine-grained pseudolabels by

modeling accuracies and correlations of label sources. The second step is training or fine-tuning the end model with pseudolabels to yield better generalization than the label model. Not only can our method improve performance and fairness, but it is also compatible with the previous WS label models, as it works at the label source level—prior to the label model.

Another related line of work is WS using embeddings of inputs [LVS, CFA$^+$22, ZSR23]. The first work, [LVS], uses embeddings for subset selection, where high-confidence subsets are selected based on the proximity to the same pseudolabels. The second, [CFA$^+$22], exploits embeddings to estimate local accuracy and improve coverage. The third [ZSR23] also incorporates instance features into label model via Gaussian process and Bayesian mixture models. Similarly, our method uses embeddings to improve fairness by mapping points in different groups in the embedded space. Embedding spaces formed from a pre-trained model typically offer better distance metrics than the input space, which provides an advantage when modeling the relationships between input points.

**Fairness in machine learning**    Fairness in machine learning is an active research area to detect and address biases in ML algorithms. There have been many suggested fairness (bias) notions and solutions for them [DHP$^+$12, HPS16, KLRS17, HNG19, GNRV19, HWZW20, GZS$^+$22]. Popular approaches include adopting constrained optimization or regularization with fairness metrics [DHP$^+$12, HPS16, ABD$^+$18, HNG19, GNRV19, HWZW20] and postprocessing model outputs to guarantee fairness notions [MW18, ZDC22]. While they have been successful in reducing bias, those methods often have a fundamental tradeoff between accuracy and fairness.

A more recent line of works using Optimal Transport (OT) [GDBFL19, BYF20, SRT$^+$20, SMBN21, BDB22] has shown that it is possible to improve fairness without such tradeoffs by uniformizing distributions in different groups. Our method improves fairness and performance simultaneously by using OT. However, OT-based methods can suffer from additional errors induced by the optimal transport mapping. In our method, OT-induced errors that occur at the label sources can be covered by the label model, since those labeling functions that are made worse are down-weighted in label model fitting. A limitation of our method is that its resulting fairness scores are unpredictable in advance, while methods having fairness-accuracy tradeoffs typically come with tunable parameters to a priori control this tradeoff. However, as we argued in Section 5, our method is compatible with such methods, so that it is possible to control the fairness of the end model by adopting other methods as well.

Another related line of research is fairness under noisy labels [WLL21, KL22, WZN$^+$23, ZZL$^+$23]. [WLL21, WGH$^+$22] suggests a new loss function that considers the noise level in labels to improve fairness. [KL22] studies PAC learning under a noisy label setting with fairness considerations. [WZN$^+$23] introduces a regularizer to boost fairness under label noise, focusing on the performance difference in subpopulations. Finally, [ZZL$^+$23] also proposes a regularizer based on mutual information to improve fairness of representations under label noise. While our method shares the assumption of label noise, the main difference is that we have access to multiple noisy estimates—whose noise rates we can learn—and where we can apply fairness notions to each separate estimate. In contrast, other methods consider only one noisy label source—which corresponds to pseudolabels generated by label models in weak supervision. Thus, their methods can be used additionally with SBM, like other previous fair ML methods such as optimal threshold. Also, the typical weak supervision pipeline does not include training for individual label sources — so that techniques involving optimizing loss functions and regularization do not fit this setting.

**Data slice discovery**    Data slice denotes a group of data examples that share a common attribute or characteristic [EVS$^+$22]. Slice discovery aims to discover semantically meaningful subgroups from unstructured input data. Typical usage of slice discovery is to identify data slices where the model underperforms. For example, [DVMWVdM19] analyzed off-the-shelf object recognition models' performance with additional context annotations, revealing that model performance can vary depending on the geographical location where the image is taken. [WFT$^+$19] shows that the melanoma detection models do not work well for dermascopic images with skin markings since they use those markings as a spurious feature. [ORDCR20] found pneumothorax classification models in chest X-rays underperform for chest x-ray images with chest drains. To discover such data slices without additional annotations, a variety of strategies have been employed. [ORDCR20, SDA$^+$20, EVS$^+$22] used clustering or mixture modeling with dimensionality reduction such as U-MAP [SDA$^+$20], SVD [EVS$^+$22]. [ddWLB22] solves an optimization problem to find high loss data points given a subgroup size.

While typical slice discovery methods require access to true labels to find underperforming data slices, weak superivison setting does not allow it. To work this around, we replace true labels with pseudolabels generated by label model. And, slice discovery methods are applied for outputs of each LF. Thus, the discovered data slices represent data points where each LF disagrees with the aggregated pseudolabels.

## C  Algorithm details

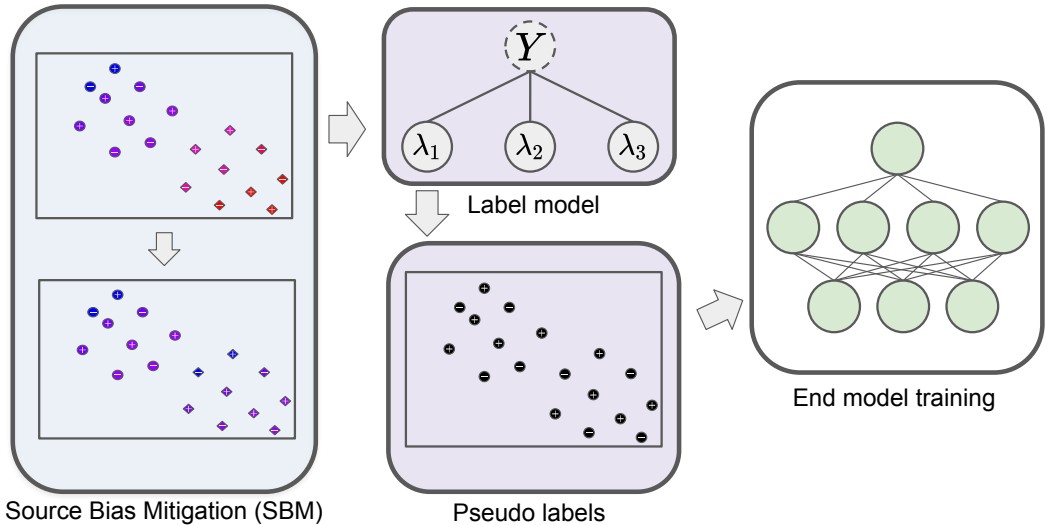

Figure 5: Overall WS pipeline involving SBM

---

**Algorithm 2:** ESTIMATE ACCURACY (TRIPLET) [FCS$^+$20]

---

**Parameters:** Weak label values $\lambda^1,...,\lambda^m$
**for** $i,j,k \in \{1,2,...,m\}$ $(i \neq j \neq k)$ **do**

$$|\hat{a}^i| \leftarrow \sqrt{\hat{\mathbb{E}}[\lambda^i \lambda^j]\hat{\mathbb{E}}[\lambda^i \lambda^k]/\hat{\mathbb{E}}[\lambda^j \lambda^k]}$$

$$|\hat{a}^j| \leftarrow \sqrt{\hat{\mathbb{E}}[\lambda^i \lambda^j]\hat{\mathbb{E}}[\lambda^j \lambda^k]/\hat{\mathbb{E}}[\lambda^i \lambda^k]}$$

$$|\hat{a}^k| \leftarrow \sqrt{\hat{\mathbb{E}}[\lambda^i \lambda^k]\hat{\mathbb{E}}[\lambda^j \lambda^k]/\hat{\mathbb{E}}[\lambda^i \lambda^j]}$$

**end for**
**return** ResolveSign $(\hat{a}^i)$     $\forall i \in \{1,2,...,m\}$

---

**Algorithm 3:** TRANSPORT

---

**Parameters:** Source input $X_{src}$, destination input $X_{dst}$, source weak label values $\lambda_{src}$, destination weak label values $\lambda_{dst}$, Optimal Transport type $O$, the number of nearest neighbors $k=1$
**if** OT type $O$ is linear **then**
    $\tilde{X}_{src} \leftarrow$ LINEAROT$(X_{src},X_{dst})$ [KS84]
**else if** OT type $O$ is sinkhorn **then**
    $\tilde{X}_{src} \leftarrow$ SINKHORNOT$(X_{src},X_{dst})$ [Cut13]
**else**
    $\tilde{X}_{src} \leftarrow X_{src}$
**end if**
$\tilde{\lambda}_{src} \leftarrow kNN(\tilde{X}_{src},X_{dst},\lambda_{dst},k)$
**return** $\tilde{\lambda}_{src}$

---

In this appendix section, we discuss the details of the algorithms we used. The overall WS pipeline including SBM is illustrated in Figure 5. Our method is placed in the first step as a refinement of

**Algorithm 4:** LINEAROT (OT-L)[KS84]

**Parameters:** Source input $X_{src}$, destination input $X_{dst}$

$\mu_s, \mu_t \leftarrow \text{MEAN}(X_{src}), \text{MEAN}(X_{dst})$
$\Sigma_s, \Sigma_t \leftarrow \text{COV}(X_{src}), \text{COV}(X_{dst})$
$A \leftarrow \Sigma_s^{-1/2}(\Sigma_s^{1/2}\Sigma_t\Sigma_s^{1/2})^{1/2}\Sigma_s^{-1/2}$
$b \leftarrow \mu_t - A\mu_s$
$n_{src} \leftarrow \text{size}(X_{src})$
**for** $i \in \{1, 2, ..., n_{src}\}$ **do**
    $\tilde{X}_{src}[i] \leftarrow AX_{src}[i] + b$
**end for**
**return** $\tilde{X}_{src}$

---

**Algorithm 5:** SINKHORNOT (OT-S)[Cut13]

**Parameters:** Source input $X_{src}$, destination input $X_{dst}$, Entropic regularization parameter $\eta = 1$

$M \leftarrow \text{PAIRWISEDISTANCE}(X_{src}, X_{dst})$
$n_{src}, n_{dst} \leftarrow \text{size}(X_{src}), \text{size}(X_{dst})$
$a \leftarrow (1/n_{src}, ..., 1/n_{src})^T \in \mathbb{R}^{n_{src}}$
$b \leftarrow (1/n_{dst}, ..., 1/n_{dst})^T \in \mathbb{R}^{n_{dst}}$
Set $K \in \mathbb{R}^{n_{src} \times n_{dst}}$ such that $K_{ij} = \exp(-M_{ij}/\eta)$
Initialize $v = (1, ..., 1)^T \in \mathbb{R}^{n_{dst}}$
**while** not converged **do**

$$u \leftarrow \frac{a}{Kv}$$
$$v \leftarrow \frac{b}{K^T u}$$

**end while**
$\pi \leftarrow \text{diag}(u)K\text{diag}(v)$
$\tilde{\pi} = n_{src}\pi$
**for** $i \in \{1, 2, ..., n_{src}\}$ **do**
    $\tilde{X}_{src}[i] \leftarrow \sum_{j=1}^{n_{dst}} \tilde{\pi}_{ij} X_{dst}[j]$
**end for**
**return** $\tilde{X}_{src}$

---

noisy labels. In this step, we improve the noisy labels by transporting the low accuracy group to the high accuracy group. To identify low accuracy and high accuracy groups, we estimate accuracy using Algorithm 2. After estimating accuracies, we transport the low accuracy group to the high accuracy group and get refined labels for the low accuracy group by Algorithm 3. Linear OT (Optimal Transport) is used to estimate a mapping. Sinkhorn OT approximates optimal coupling using Sinkhorn iteration. After obtaining the optimal coupling, data points from the low accuracy group are mapped to the coupled data points in the high accuracy group. If OT type is not given, data points from the low accuracy group are mapped to the 1-nearest neighbor in the high accuracy group. After obtaining mapped points, weak labels associated with mapped points in the target group are used for the data points from the low accuracy group.

The next step is a standard weak supervision pipeline step – training a label model and running inference to obtain pseudolabels. Finally, we train the end model. We used the off-the-shelf label model Snorkel [BRL+19] for the label model and logistic regression as the end model in all experiments unless otherwise stated. For the optimal transport algorithm implementation, we used the Python POT package [FCG+21].

## D Theory details

Accuracy is defined as $P(\lambda(x) = y) = P(\lambda(x) = 1, y = 1) + P(\lambda(x) = -1, y = -1)$. For ease of notation, below we set $\phi(x, x^{\text{center}_0}) = (1 + \|x - x^{\text{center}_0}\|)^{-1}$.

## D.1 Proof of Theorem 4.1

**Theorem 4.1.** *Let $g_1^{(k)}$ be an arbitrary sequence of functions such that $\lim_{k\to\infty}\mathbb{E}_{x'\in g_1^{(k)}(\mathcal{X})}[\|x'-x^{center_0}\|] \to \infty$. Suppose our assumptions above are met; in particular, that the label $y$ is independent of the observed features $x=I(z)$ or $x'=g_1^{(k)}(z),\forall k$, conditioned on the latent features $z$. Then,*

$$\lim_{k\to\infty}\mathbb{E}_{x'\in g_1^{(k)}(\mathcal{Z})}[P(\lambda(x')=y)]=\frac{1}{2},$$

*which corresponds to random guessing.*

*Proof of Theorem 4.1.* Because $\lim_{k\to\infty}\mathbb{E}_{x'\in g_1^{(k)}(\mathcal{Z})}[\|x'-x^{center_0}\|]\to\infty$, we have

$$\lim_{k\to\infty}\mathbb{E}_{x'\in g_1^{(k)}(\mathcal{Z})}[\phi(x',x^{center_0})]=0$$

and because $Z=\sum_{i\in\{0,1\}}\sum_{j\in\{0,1\}}P(\lambda(x')=i,y=j)$,

$$\lim_{k\to\infty}\mathbb{E}_{x'\in g_1^{(k)}(\mathcal{Z})}[P(\lambda(x')=y)]=\lim_{k\to\infty}\mathbb{E}_{x'\in g_1^{(k)}(\mathcal{Z})}[P(\lambda(x')=1,y=1)+P(\lambda(x')=-1,y=-1)]$$

$$=\lim_{k\to\infty}\mathbb{E}_{x'\in g_1^{(k)}(\mathcal{Z})}\left[\frac{1}{Z}\exp(\theta_0(1)(1)\phi(x',x^{center_0}))+\frac{1}{Z}\exp(\theta_0(-1)(-1)\phi(x',x^{center_0}))\right]$$

$$=\lim_{k\to\infty}\mathbb{E}_{x'\in g_1^{(k)}(\mathcal{Z})}\left[\frac{2}{Z}\exp(\theta_0\phi(x',x^{center_0}))\right]$$

$$=\lim_{k\to\infty}\mathbb{E}_{x'\in g_1^{(k)}(\mathcal{Z})}\left[\frac{2\exp(\theta_0\phi(x',x^{center_0}))}{2(\exp(\theta_0\phi(x',x^{center_0}))+\exp(-\theta_0\phi(x',x^{center_0})))}\right]$$

$$=\lim_{k\to\infty}\mathbb{E}_{x'\in g_1^{(k)}(\mathcal{Z})}\left[\frac{\exp(\theta_0\phi(x',x^{center_0}))}{\exp(\theta_0\phi(x',x^{center_0}))+\exp(-\theta_0\phi(x',x^{center_0}))}\right]$$

$$=\frac{1}{2}.$$

since we get

$$\lim_{k\to\infty}\mathbb{E}_{x'\in g_1^{(k)}(\mathcal{Z})}\exp(\theta_0\phi(x',x^{center_0}))=1$$

$$\lim_{k\to\infty}\mathbb{E}_{x'\in g_1^{(k)}(\mathcal{Z})}\exp(-\theta_0\phi(x',x^{center_0}))=1$$

from $\lim_{k\to\infty}\mathbb{E}_{x'\in g_1^{(k)}(\mathcal{Z})}[\phi(x',x^{center_0})]=0$ ▢

## D.2 Proof of Theorem 4.2

**Theorem 4.2.** *Let $\tau = \max\left(\frac{\mathbf{r}(\Sigma_0)}{n_0},\frac{\mathbf{r}(\Sigma_1)}{n_1},\frac{t}{\min(n_0,n_1)},\frac{t^2}{\max(n_0,n_1)^2}\right)$ and $C$ be a constant. Using Algorithm 1, for any $t>0$, we bound the difference*

$$|\mathbb{E}_{x\in\mathcal{Z}}[P(\lambda(x)=y)]-\mathbb{E}_{x'\in\mathcal{X}}[P(\lambda(\hat{h}(x'))=y)]|$$
$$\leq 4\theta_0 C\sqrt{\tau\mathbf{r}(\Sigma_1)}.$$

*with probability $1-e^{-t}-\frac{1}{n_1}$.*

We use the result from [FLF19], which bounds the difference of the true $h$ and empirical $\hat{h}$ Monge estimators between two distributions $P_0(x)$ and $P_1(x')$. Let $\mathbf{r}(\Sigma)$, $\lambda_{\min}(\Sigma)$ and $\lambda_{\max}(\Sigma)$ denote the effective rank, minimum and maximum eigenvalues of matrix $\Sigma$ respectively. Then,

**Lemma D.1** ([FLF19]). *Let $P_0(x)$ and $P_1(x')$ be sub-Gaussian distributions on $\mathcal{X}=\mathbb{R}^d$ with expectations $\mu_0,\mu_1$ and positive-definite covariance operators $\Sigma_0,\Sigma_1$ respectively. We observe $n_0$ points from the distribution $P_0(x)$ and $n_1$ points from the distribution $P_1(x')$. We assume that*

$$c_1<\min_{j\in\{0,1\}}\lambda_{\min}(\Sigma_j)\leq\max_{j\in\{0,1\}}\lambda_{\max}(\Sigma_j)\leq c_2,$$

*for fixed constants $0 < c_1 \leq c_2 < \infty$. Further, we assume $n_0 \geq c\mathbf{r}(\Sigma_0)$ and $n_1 \geq d$ for a sufficiently large constant $c > 0$.*

*Then, for any $t > 0$, we have with probability at least $1 - e^{-t} - \frac{1}{n_1}$ that*

$$\mathbb{E}_{x' \in \mathcal{X}}\left[\left\|h(x') - \hat{h}(x')\right\|\right] \leq C\sqrt{\tau \mathbf{r}(\Sigma_1)},$$

*where $C$ is a constant independent of $n_0$, $n_1$, $d$, $\mathbf{r}(\Sigma_0)$, $\mathbf{r}(\Sigma_1)$ and $\tau = \max\left(\frac{\mathbf{r}(\Sigma_0)}{n_0}, \frac{\mathbf{r}(\Sigma_1)}{n_1}, \frac{t}{\min(n_0, n_1)}, \frac{t^2}{\max(n_0, n_1)^2}\right)$.*

We will also use the following:

**Lemma D.2.** *The probability $P(\lambda(x) = y)$ is $L$-Lipschitz with respect to $x \in \mathcal{X}$. Specifically, $\forall x_1, x_2 \in \mathcal{X}$,*
$$|P(\lambda(x_1) = y) - P(\lambda(x_2) = y)| \leq 4\theta_0 \|x_1 - x_2\|.$$

*Proof.* We will demonstrate that, because $\|\nabla_x P(\lambda(x) = y)\|$ is bounded above by $4\theta_0$, $P(\lambda(x) = y)$ must be $4\theta_0$-Lipschitsz with respect to $x$. First,

$$P(\lambda(x) = y) = \frac{\exp(\theta_0 \phi(x, x^{\text{center}_0}))}{\exp(\theta_0 \phi(x, x^{\text{center}_0})) + \exp(-\theta_0 \phi(x, x^{\text{center}_0}))}$$
$$= \frac{1}{1 + \exp(-2\theta_0 \phi(x, x^{\text{center}_0}))}$$
$$= \sigma(2\theta_0 \phi(x, x^{\text{center}_0})),$$

where $\sigma$ denotes the sigmoid function. Note that $\frac{d}{du}\sigma(u) = \frac{\exp(-u)}{(1+\exp(-u))^2}$ for $u \in \mathbb{R}$ and that $\nabla_x[2\theta_0 \phi(x, x^{\text{center}_0})] = 2\theta_0 \nabla_x\left[\frac{1}{1 + \|x - x^{\text{center}_0}\|}\right] = 2\theta_0 \frac{\nabla_x[\|x - x^{\text{center}_0}\|]}{(1 + \|x - x^{\text{center}_0}\|)^2}$. Further, note that $\nabla_x[\|x - x^{\text{center}_0}\|] = \frac{x - x^{\text{center}_0}}{\|x - x^{\text{center}_0}\|}$ because we use Euclidean norm. Thus,

$$\|\nabla_x P(\lambda(x) = y)\| = \left\|\nabla_x \sigma(2\theta_0 \phi(x, x^{\text{center}_0}))\right\|$$
$$= \left\|2\theta_0 \frac{exp(-2\theta_0 \phi(x, x^{\text{center}_0}))}{(1 + exp(-2\theta_0 \phi(x, x^{\text{center}_0})))^2 (1 + \|x - x^{\text{center}_0}\|)^2} \cdot \nabla_x[\|x - x^{\text{center}_0}\|]\right\|$$
$$= \left\|2\theta_0 \frac{exp(-2\theta_0 \phi(x, x^{\text{center}_0}))}{(1 + exp(-2\theta_0 \phi(x, x^{\text{center}_0})))^2} \phi(x, x^{\text{center}_0})^2 \cdot \frac{x - x^{\text{center}_0}}{\|x - x^{\text{center}_0}\|}\right\|$$
$$= \left\|2\theta_0 \sigma(2\theta_0 \phi(x, x^{\text{center}_0}))(1 - \sigma(2\theta_0 \phi(x, x^{\text{center}_0})))\phi(x, x^{\text{center}_0})^2 \cdot \frac{x - x^{\text{center}_0}}{\|x - x^{\text{center}_0}\|}\right\|$$
$$< 2\theta_0 \left\|1 \cdot (1 - (-1))\phi(x, x^{\text{center}_0})^2 \cdot \frac{x - x^{\text{center}_0}}{\|x - x^{\text{center}_0}\|}\right\|$$
$$= 4\theta_0 \left\|\phi(x, x^{\text{center}_0})^2 \cdot 1\right\|$$
$$\leq 4\theta_0.$$

Thus, $\|\nabla_x P(\lambda(x) = y)\|$ is bounded above by $4\theta_0$. We now use this fact to demonstrate that $P(\lambda(x) = y)$ is $4\theta_0$-Lipschitz with respect to $x$.

For $0 \leq v \leq 1$, let $s(v) = x_1 + (x_2 - x_1)v$. For $x$ between $x_1$ and $x_2$ inclusive and for $v$ such that $s(v) = x$, since $\|\nabla_x P(\lambda(x) = y)\| = |\frac{d}{dv}P(\lambda(s(v)) = y)| \leq 4\theta_0$, we have

$$|P(\lambda(x_1) = y) - P(\lambda(x_2) = y)| = \left|\int_{v=0}^{1}\left[\frac{d}{dv}P(\lambda(s(v)) = y)\right]\|s(v)\|dv\right|$$
$$\leq \int_{v=0}^{1}\left|\frac{d}{dv}P(\lambda(s(v)) = y)\right|\|s(v)\|dv$$
$$\leq \int_{v=0}^{1} 4\theta_0 \|s(v)\|dv$$
$$= 4\theta_0 \|x_1 - x_2\|.$$

$\square$

*Proof of Theorem 4.2.* Now we are ready to complete our proof. We have,

$$
\begin{aligned}
|\mathbb{E}_{x\in\mathcal{Z}}&[P(\lambda(x)\!=\!y)]\!-\!\mathbb{E}_{x'\in\mathcal{X}}[P(\lambda(\hat{h}(x')\!)\!=\!y)]| \\
&\leq \mathbb{E}_{z\in\mathcal{Z}}[|P(\lambda(I(z))\!=\!y)\!-\!P(\lambda(\hat{h}(g_1(z)))\!=\!y|] \\
&= \mathbb{E}_{z\in\mathcal{Z}}[|P(\lambda(h(g_1(z)))\!=\!y)\!-\!P(\lambda(\hat{h}(g_1(z)))\!=\!y|] \\
&\leq \mathbb{E}_{z\in\mathcal{Z}}[4\theta_0||h(g_1(z))\!-\!\hat{h}(g_1(z))||] \text{ (by Lemma D.2)} \\
&= \mathbb{E}_{x'\in\mathcal{X}}[4\theta_0||h(x')\!-\!\hat{h}(x')||] \\
&\leq 4\theta_0 C\sqrt{\tau\mathbf{r}(\Sigma_1)},
\end{aligned}
$$

where the last line holds with probability at least $1\!-\!e^{-t}\!-\!\frac{1}{n_1}$ by Lemma D.1.

$\square$

# E   Experiment details

## E.1   Dataset details

Table 7: Summary of real datasets

| Domain | Datasets | Y | A | $P(Y\!=\!1\|A\!=\!1)$ | $P(Y\!=\!1\|A\!=\!0)$ | $\Delta_{DP}$ |
|---|---|---|---|---|---|---|
| Tabular | Adult | Income ($\geq 50000$) | Sex | 0.3038 | 0.1093 | 0.1945 |
| | Bank | Subscription | Age ($\geq 25$) | 0.1093 | 0.2399 | 0.1306 |
| NLP | CivilComments | Toxicity | Black | 0.3154 | 0.1057 | 0.2097 |
| | HateXplain | Toxicity | Race | 0.8040 | 0.4672 | 0.3368 |
| Vision | CelebA | Gender | Young | 0.3244 | 0.6590 | 0.3346 |
| | UTKFace | Gender | Asian | 0.4856 | 0.4610 | 0.0246 |

**Adult**   The adult dataset [K+96] has information about the annual income of people and their demographics. The classification task is to predict whether the annual income of a person exceeds \$50,000 based on demographic features. Demographic features include age, work class, education, marital status, occupation, relationship, race, capital gain and loss, work hours per week, and native country. The group variable is whether age is greater than 25 or not. The training data has 32,561 examples, and the test data has 16,281 examples.

**BankMarketing**   The bank marketing dataset [MCR14] was collected during the direct marketing campaigns of a Portuguese banking institution from 2008 to 2013. Features consist of demographic attributes, financial attributes, and the attributes related to the interaction with the campaign. The task is to classify whether a client will make a deposit subscription or not. The group variable is, whether age is greater than 25 or not. The total number of instances is 41,188, where the training dataset has 28,831 rows and the test dataset has 12,357 rows.

**CivilComments**   The CivilComments dataset [BDS+19] is constructed from data collected in Jigsaw online conversattion platform. We used the dataset downloaded from [KSM+21]. Features are extracted using BERT [DCLT18] embeddings from comments, and the task is to classify whether a given comment is toxic or not. This dataset has annotations about the content type of comments, e.g. race, age, religion, etc. We chose "black" as the group variable. The total number of instances is 402,820, where the training dataset has 269,038 rows and the test dataset has 133,782 rows.

**HateXplain**   The HateXplain dataset [MSY+21] is comments collected from Twitter and Gab and labeled through Amazon Mechanical Turk. Features are extracted using BERT embeddings from comments. The task is to classify whether a given comment is toxic or not. This dataset has three label types - hate, offensive, or normal. We used hate and offensive label types as "toxic" label, that is, if

one of two labels types is 1 for a row, the row has label 1. The dataset has annotations about the target communities - race, religion, gender, LGBTQ. We used the race group as the group variable. The total number of instances is 17,281, where the training set has 15,360 rows and the test set has 1,921 rows.

**CelebA**  The CelebA dataset [LLWT15] is the image datasets that has images of faces with various attributes. Features are extracted using CLIP[RKH$^+$21] embeddings from images. We tried gender classification task from this dataset—various other tasks can be constructed. We used "Young" as the group variable. The total number of instances is 202,599, where the training set has 162,770 rows and the test set has 39,829 rows.

**UTKFace**  UTKFace dataset [ZSQ17] consists of faces images in the wild with annotations of age, gender, and ethnicity. Features are extracted using CLIP embeddings from images. This dataset has attributes of age, gender, race and we chose gender as label (Male:0, Female: 1), solving gender classification. We used the race as the group variable, encoding "Asian" group as 1, other race groups as 0. The total number of instances is 23,705, where the training set has 18,964 rows and the test set has 4,741 rows.

## E.2   Application of LIFT

To use smooth embeddings in the transport step of tabular datasets (adult and bank marketing), we convert tabular data rows into text sentences and then get BERT embeddings.

**Adult**  First we decided r.pronoun for each row r from <r.sex> and then generated sentences with the following template. "<r.pronoun> is in her <r.age_range>s. <r.pronoun> is from <r.nationality>. <r.pronoun> works in <r.workclass>. Specifically, <r.pronoun> has a <r.job> job. <r.pronoun> works <r.working_hour> hours per week. <r.pronoun> total education year is <r.education_years>. <r.pronoun> is <r.marital_status>. <r.pronoun> is <r.race>".

An example sentence is "She is in her 20s. She is from United-States. She works in private sector. Specifically, she has a sales job. She works 30 hours per week. Her total education year is 6 years. She is married. She is White."

**Bank marketing**  Similarly, text sentences for each row r are generated using the following template. "This person is <r.age> years old. This person works in a <r.job> job. This person is <r.marital_status>. This person has a <r.education_degree> degree. This person <if r.has_housing_loan "has" else "doesn't have"> a housing loan. This person <if r.has_personal_loan "has" else "doesn't have"> a personal loan. THis person's contact is <r.contact_type>. The last contact was on <r.last_contact_weekday> in <r.last_contact_month>. The call lasted for <r.duration> seconds. <r.campaign> times of contacts performed during this campaign. Before this campaign, <r.previous> times of contacts have been performed for this person. The employment variation rate of this person is <r.emp_var_rate>%. The consumer price index of this person is <r.cons_price_idx>%. The consumer confidence index of this person is <r.cons_conf_idx>%. The euribor 3 month rate of this person is <r.euribor3m>. The number of employees is <r.nr_employed>".

An example sentence is "This person is 37 years old. This person works in a management job. This person is married. This person has a university degree. This person doesn't have a housing loan. This person has a personal loan. This person's contact is a cellular. The last contact was on a Thursday in May. The call lasted for 195 seconds. 1 times of contacts performed during this campaign. Before this campaign, 0 times of contacts have been performed for this person. The employment variation rate of this person is -1.8%. The consumer price index of this person is 92.893%. The consumer confidence index of this person is -46.2%. The euribor 3 month rate of this person is 1.327%. The number of employees is 5099."

## E.3   LF details

In this subsection, we describe the labeling functions used in the experiments. The performances of labeling functions, including our SBM results, are reported in Table 8 ∼ 19. In SBM result tables, performance, fairness improvement by SBM are green colored.

**Adult**    We generated heuristic labeling functions based on the features as follows.

- LF 1 (age LF): True if the age is between 30 and 60. False otherwise.
- LF 2 (education LF): True if the person has a Bachelor, Master, PhD degree. False otherwise
- LF 3 (marital LF): True if marital status is married. False otherwise.
- LF 4 (relationship LF): True if relationship status is Wife, Own-child, or Husband, False otherwise.
- LF 5 (capital LF): True if the capital gain is >5000. False otherwise.
- LF 6 (race LF): True if Asian-Pac-Islander or race-Other. False otherwise.
- LF 7 (country LF): True if Germany, Japan, Greece, or China. False otherwise.
- LF 8 (workclass LF): True if the workclass is Self employed, federal government, local government, or state government. False otherwise.
- LF 9 (occupation LF): True if the occupation is sales, executive managerial, professional, or machine operator. False otherwise.

**Bank marketing**    Similar to Adult dataset, we generated heuristic labeling functions based on the features as follows.

- LF 1 (loan LF): True if the person has loan previously.False otherwise.
- LF 2 (previous contact LF): True if the previous contact number is >1.1. False otherwise.
- LF 3 (duration LF): True if the duration of bank marketing phone call is > 6 min.  False otherwise.
- LF 4 (marital LF): True if marital status is single. False otherwise.
- LF 5 (previous outcome LF): True if the previous campaign was successful. False otherwise.
- LF 6 (education LF): True if the education level is university degree or professional course taken. False otherwise.

**CivilComments**    We generated heuristic labeling functions based on the inclusion of specific word lists. If a comment has a word that is included in the word list of LF, it gets True for that LF.

- LF 1 (sex LF): ["man", "men", "woman", "women", "male", "female", "guy", "boy", "girl", "daughter", "sex", "gender", "husband", "wife", "father", "mother", "rape", "mr.", "feminist", "feminism", "pregnant", "misogy", "pussy", "penis", "vagina", "butt", "dick"]
- LF 2 (LGBTQ LF): ["gay", "lesbian", "transgender", "homosexual", "homophobic", "heterosexual", "anti-gay", "same-sex", "bisexual", "biological"]
- LF 3 (religion LF): ["muslim", "catholic", "church", "christian",  "god", "jesus", "christ", "jew", "islam", "israel", "bible", "bishop", "gospel", "clergy", "protestant", "islam"]
- LF 4 (race LF): ["white", "black", "racist", "trump", "supremacist", "american", "canada", "kkk", "nazi", "facist", "african", "non-white", "discrimination", "hate", "neo-nazi", "asia", "china", "chinese"]
- LF 5 (toxic LF): ["crap", "damn", "bitch", "ass", "fuck", "bullshit", "hell", "jerk"]
- LF 6 (threat LF): ["shoot", "kill", "shot", "burn", "stab", "murder", "gun", "fire", "rape", "punch", "hurt", "hunt", "bullet", "hammer"]
- LF 7 (insult LF): ["stupid", "idiot", "dumb", "liar", "poor", "disgusting", "moron", "nasty", "lack", "brain", "incompetent", "sociopath"]

**HateXplain**    We used heuristic the same labeling functions with CivilComments. However, their performance was close to random guess (accuracy close to 0.5), so we added 5 pretrained model LFs from detoxify [HU20] repository.  We used models listed as original, unbiased, multilingual, original-small, unbiased-small.

**CelebA**    We used models pretrained from other datasets as LFs.

- LF 1: ResNet-18 fine-tuned on gender classification dataset [1]
- LF 2: ResNet-34 fine-tuned on FairFace dataset
- LF 3: ResNet-34 fine-tuned on UTKFace dataset
- LF 4: ResNet-34 fine-tuned on UTKFace dataset (White only)
- LF 5: ResNet-34 fine-tuned on UTKFace dataset (non-White only)
- LF 6: ResNet-34 fine-tuned on UTKFace dataset (age $\geq 50$ only)

---

[1]https://www.kaggle.com/datasets/cashutosh/gender-classification-dataset

Table 8: Tabular dataset raw LF performance

| Dataset | LF | Acc | F1 | $\Delta_{DP}$ | $\Delta_{EO}$ |
|---|---|---|---|---|---|
| Adult | LF 1 (age LF) | 0.549 | 0.476 | 0.100 | 0.023 |
| | LF 2 (education LF) | 0.743 | 0.455 | 0.033 | 0.044 |
| | LF 3 (marital LF) | 0.699 | 0.579 | 0.447 | 0.241 |
| | LF 4 (relationship LF) | 0.564 | 0.486 | 0.381 | 0.243 |
| | LF 5 (capital LF) | 0.800 | 0.315 | 0.035 | 0.019 |
| | LF 6 (race LF) | 0.737 | 0.066 | 0.003 | 0.004 |
| | LF 7 (country LF) | 0.756 | 0.024 | 0.001 | 0.004 |
| | LF 8 (workclass LF) | 0.678 | 0.399 | 0.066 | 0.003 |
| | LF 9 (occupation LF) | 0.644 | 0.466 | 0.013 | 0.012 |
| Bank Marketing | LF 1 (loan LF) | 0.769 | 0.124 | 0.004 | 0.013 |
| | LF 2 (previous contact LF) | 0.888 | 0.187 | 0.055 | 0.068 |
| | LF 3 (duration LF) | 0.815 | 0.415 | 0.019 | 0.125 |
| | LF 4 (marital LF): | 0.682 | 0.197 | 0.602 | 0.643 |
| | LF 5 (previous outcome LF) | 0.898 | 0.301 | 0.052 | 0.062 |
| | LF 6 (education LF) | 0.575 | 0.206 | 0.183 | 0.219 |

- LF 7: ResNet-34 fine-tuned on UTKFace dataset (age $< 50$ only)

**UTKFace**   We used models pretrained from other datasets as LFs.

- LF 1: ResNet-18 fine-tuned on gender classification dataset
- LF 2: ResNet-34 fine-tuned on gender classification dataset
- LF 3: ResNet-34 fine-tuned on CelebA dataset
- LF 4: ResNet-34 fine-tuned on CelebA dataset (attractive only)
- LF 5: ResNet-34 fine-tuned on CelebA dataset (non-attractive only)
- LF 6: ResNet-34 fine-tuned on CelebA dataset (young only)
- LF 7: ResNet-34 fine-tuned on CelebA dataset (non-young only)
- LF 8: ResNet-34 fine-tuned on CelebA dataset (unfair sampling)

Table 9: Tabular dataset SBM (w/o OT) LF performance. $\Delta$s are obtained by comparison with raw LF performance.

| Dataset | LF | Acc ($\Delta$) | F1 ($\Delta$) | $\Delta_{DP}$ ($\Delta$) | $\Delta_{EO}$ ($\Delta$) |
|---|---|---|---|---|---|
| Adult | LF 1 | 0.597 (0.048) | 0.594 (0.118) | 0.119 (0.019) | 0.106 (0.083) |
| | LF 2 | 0.601 (-0.142) | 0.346 (-0.109) | 0.035 (0.002) | 0.019 (-0.025) |
| | LF 3 | 0.998 (0.299) | 0.998 (0.419) | 0.303 (-0.144) | 0.001 (-0.240) |
| | LF 4 | 0.719 (0.155) | 0.723 (0.237) | 0.304 (-0.077) | 0.091 (-0.152) |
| | LF 5 | 0.649 (-0.151) | 0.183 (-0.132) | 0.035 (0.000) | 0.023 (0.004) |
| | LF 6 | 0.615 (-0.122) | 0.082 (0.016) | 0.003 (0.000) | 0.029 (0.025) |
| | LF 7 | 0.621 (-0.135) | 0.023 (-0.001) | 0.001 (0.000) | 0.005 (0.001) |
| | LF 8 | 0.612 (-0.066) | 0.318 (-0.021) | 0.019 (-0.047) | 0.027 (0.024) |
| | LF 9 | 0.565 (-0.079) | 0.460 (-0.006) | 0.013 (0.000) | 0.026 (0.014) |
| Adult (LIFT) | LF 1 | 0.506 (-0.043) | 0.415 (-0.061) | 0.309 (0.209) | 0.198 (0.175) |
| | LF 2 | 0.861 (0.118) | 0.691 (0.236) | 0.080 (0.047) | 0.126 (0.082) |
| | LF 3 | 0.691 (-0.008) | 0.188 (-0.391) | 0.130 (-0.317) | 0.186 (-0.055) |
| | LF 4 | 0.391 (-0.173) | 0.307 (-0.179) | 0.455 (0.074) | 0.314 (0.071) |
| | LF 5 | 0.725 (-0.075) | 0.117 (-0.198) | 0.014 (-0.021) | 0.029 (0.010) |
| | LF 6 | 0.703 (-0.034) | 0.109 (0.043) | 0.003 (0.000) | 0.004 (0.000) |
| | LF 7 | 0.708 (-0.048) | 0.036 (0.012) | 0.001 (0.000) | 0.000 (-0.004) |
| | LF 8 | 0.897 (0.219) | 0.800 (0.461) | 0.033 (-0.033) | 0.235 (0.232) |
| | LF 9 | 0.601 (-0.043) | 0.446 (-0.020) | 0.013 (0.000) | 0.068 (0.056) |
| Bank | LF 1 | 0.706 (-0.063) | 0.165 (0.041) | 0.004 (0.000) | 0.003 (-0.010) |
| | LF 2 | 0.824 (-0.064) | 0.221 (0.034) | 0.037 (-0.018) | 0.113 (0.045) |
| | LF 3 | 0.931 (0.116) | 0.829 (0.414) | 0.019 (0.000) | 0.110 (-0.015) |
| | LF 4 | 0.427 (-0.255) | 0.411 (0.214) | 0.100 (-0.502) | 0.004 (-0.639) |
| | LF 5 | 0.833 (-0.065) | 0.287 (-0.014) | 0.061 (0.009) | 0.195 (0.133) |
| | LF 6 | 0.624 (0.049) | 0.172 (-0.034) | 0.009 (-0.174) | 0.046 (-0.173) |
| Bank (LIFT) | LF 1 | 0.768 (-0.001) | 0.105 (-0.019) | 0.000 (-0.003) | 0.013 (0.000) |
| | LF 2 | 0.888 (0.000) | 0.188 (0.001) | 0.037 (-0.017) | 0.068 (0.000) |
| | LF 3 | 0.815 (0.000) | 0.415 (0.000) | 0.019 (0.000) | 0.125 (0.000) |
| | LF 4 | 0.317 (-0.365) | 0.228 (0.031) | 0.100 (-0.502) | 0.065 (-0.578) |
| | LF 5 | 0.898 (0.000) | 0.303 (0.001) | 0.061 (0.009) | 0.086 (0.023) |
| | LF 6 | 0.680 (0.105) | 0.126 (-0.081) | 0.009 (-0.175) | 0.084 (-0.134) |

Table 10: Tabular dataset SBM (OT-L) LF performance. $\Delta$s are obtained by comparison with raw LF performance.

| Dataset | LF | Acc ($\Delta$) | F1 ($\Delta$) | $\Delta_{DP}$ ($\Delta$) | $\Delta_{EO}$ ($\Delta$) |
|---|---|---|---|---|---|
| Adult | LF 1 | 0.841 (0.292) | 0.846 (0.370) | 0.654 (0.554) | 0.733 (0.710) |
| | LF 2 | 0.334 (-0.409) | 0.000 (-0.455) | 0.208 (0.175) | 0.000 (-0.044) |
| | LF 3 | 0.346 (-0.353) | 0.000 (-0.579) | 0.174 (-0.273) | 0.000 (-0.241) |
| | LF 4 | 0.895 (0.331) | 0.904 (0.418) | 0.735 (0.354) | 0.824 (0.581) |
| | LF 5 | 0.394 (-0.406) | 0.000 (-0.315) | 0.027 (-0.008) | 0.000 (-0.019) |
| | LF 6 | 0.408 (-0.329) | 0.070 (0.004) | 0.003 (0.000) | 0.037 (0.033) |
| | LF 7 | 0.405 (-0.351) | 0.019 (-0.005) | 0.001 (0.000) | 0.010 (0.006) |
| | LF 8 | 0.336 (-0.342) | 0.000 (-0.339) | 0.202 (0.136) | 0.000 (-0.003) |
| | LF 9 | 0.501 (-0.143) | 0.512 (0.046) | 0.013 (0.000) | 0.439 (0.427) |
| Adult (LIFT) | LF 1 | 0.628 (0.079) | 0.655 (0.179) | 0.036 (-0.064) | 0.027 (0.004) |
| | LF 2 | 0.780 (0.037) | 0.664 (0.209) | 0.014 (-0.019) | 0.032 (-0.012) |
| | LF 3 | 0.525 (-0.174) | 0.296 (-0.283) | 0.097 (-0.350) | 0.013 (-0.228) |
| | LF 4 | 0.497 (-0.067) | 0.554 (0.068) | 0.137 (-0.244) | 0.150 (-0.093) |
| | LF 5 | 0.606 (-0.194) | 0.177 (-0.138) | 0.023 (-0.012) | 0.042 (0.023) |
| | LF 6 | 0.565 (-0.172) | 0.089 (0.023) | 0.003 (0.000) | 0.008 (0.004) |
| | LF 7 | 0.567 (-0.189) | 0.029 (0.005) | 0.001 (0.000) | 0.000 (-0.004) |
| | LF 8 | 0.618 (-0.060) | 0.409 (0.070) | 0.011 (-0.055) | 0.017 (0.014) |
| | LF 9 | 0.843 (0.199) | 0.817 (0.351) | 0.013 (0.000) | 0.046 (0.034) |
| Bank | LF 1 | 0.818 (0.049) | 0.062 (-0.062) | 0.004 (0.000) | 0.140 (0.127) |
| | LF 2 | 0.980 (0.092) | 0.703 (0.516) | 0.024 (-0.031) | 0.542 (0.474) |
| | LF 3 | 0.777 (-0.038) | 0.093 (-0.322) | 0.019 (0.000) | 0.265 (0.139) |
| | LF 4 | 0.047 (-0.635) | 0.083 (-0.114) | 0.131 (-0.471) | 1.000 (0.357) |
| | LF 5 | 0.988 (0.090) | 0.839 (0.538) | 0.032 (-0.020) | 0.723 (0.660) |
| | LF 6 | 0.950 (0.375) | 0.000 (-0.206) | 0.244 (0.061) | 0.000 (-0.219) |
| Bank (LIFT) | LF 1 | 0.125 (-0.645) | 0.197 (0.073) | 0.853 (0.849) | 0.867 (0.854) |
| | LF 2 | 0.888 (0.000) | 0.180 (-0.007) | 0.000 (-0.055) | 0.036 (-0.032) |
| | LF 3 | 0.815 (0.000) | 0.415 (0.000) | 0.018 (-0.001) | 0.137 (0.012) |
| | LF 4 | 0.876 (0.194) | 0.085 (-0.112) | 0.869 (0.268) | 0.954 (0.311) |
| | LF 5 | 0.898 (0.000) | 0.300 (-0.002) | 0.047 (-0.005) | 0.039 (-0.023) |

Table 11: Tabular dataset SBM (OT-S) LF performance. $\Delta$s are obtained by comparison with raw LF performance.

| Dataset | LF | Acc ($\Delta$) | F1 ($\Delta$) | $\Delta_{DP}$ ($\Delta$) | $\Delta_{EO}$ ($\Delta$) |
|---------|------|----------------|----------------|--------------------------|--------------------------|
| Adult | LF 1 | 0.604 (0.055) | 0.595 (0.119) | 0.145 (0.045) | 0.125 (0.102) |
| | LF 2 | 0.601 (-0.142) | 0.340 (-0.115) | 0.036 (0.003) | 0.015 (-0.029) |
| | LF 3 | 0.998 (0.299) | 0.998 (0.419) | 0.293 (-0.154) | 0.002 (-0.239) |
| | LF 4 | 0.690 (0.126) | 0.700 (0.214) | 0.221 (-0.160) | 0.050 (-0.193) |
| | LF 5 | 0.656 (-0.144) | 0.181 (-0.134) | 0.032 (-0.003) | 0.021 (0.002) |
| | LF 6 | 0.621 (-0.116) | 0.081 (0.015) | 0.003 (0.000) | 0.030 (0.026) |
| | LF 7 | 0.627 (-0.129) | 0.023 (-0.001) | 0.001 (0.000) | 0.005 (0.001) |
| | LF 8 | 0.607 (-0.071) | 0.256 (-0.083) | 0.069 (0.003) | 0.095 (0.092) |
| | LF 9 | 0.558 (-0.086) | 0.446 (-0.020) | 0.013 (0.000) | 0.014 (0.002) |
| Adult (LIFT) | LF 1 | 0.475 (-0.074) | 0.436 (-0.040) | 0.031 (-0.069) | 0.032 (0.009) |
| | LF 2 | 0.953 (0.210) | 0.910 (0.455) | 0.047 (0.014) | 0.079 (0.035) |
| | LF 3 | 0.712 (0.013) | 0.191 (-0.388) | 0.157 (-0.290) | 0.250 (0.009) |
| | LF 4 | 0.373 (-0.191) | 0.425 (-0.061) | 0.204 (-0.177) | 0.120 (-0.123) |
| | LF 5 | 0.764 (-0.036) | 0.301 (-0.014) | 0.035 (0.000) | 0.107 (0.088) |
| | LF 6 | 0.711 (-0.026) | 0.117 (0.051) | 0.003 (0.000) | 0.018 (0.014) |
| | LF 7 | 0.715 (-0.041) | 0.037 (0.013) | 0.001 (0.000) | 0.002 (-0.002) |
| | LF 8 | 0.716 (0.038) | 0.272 (-0.067) | 0.147 (0.081) | 0.284 (0.281) |
| | LF 9 | 0.640 (-0.004) | 0.495 (0.029) | 0.013 (0.000) | 0.213 (0.201) |
| Bank | LF 1 | 0.694 (-0.075) | 0.173 (0.049) | 0.004 (0.000) | 0.017 (0.004) |
| | LF 2 | 0.806 (-0.082) | 0.209 (0.022) | 0.059 (0.004) | 0.211 (0.143) |
| | LF 3 | 0.949 (0.134) | 0.881 (0.466) | 0.019 (0.000) | 0.136 (0.011) |
| | LF 4 | 0.351 (-0.331) | 0.382 (0.185) | 0.040 (-0.562) | 0.016 (-0.627) |
| | LF 5 | 0.814 (-0.084) | 0.267 (-0.034) | 0.069 (0.017) | 0.247 (0.185) |
| | LF 6 | 0.583 (0.008) | 0.020 (-0.186) | 0.040 (-0.143) | 0.103 (-0.116) |
| Bank (LIFT) | LF 1 | 0.769(-0.001) | 0.087 (-0.038) | 0.006 (0.003) | 0.038 (0.025) |
| | LF 2 | 0.888 (0.001) | 0.204 (0.017) | 0.141 (0.087) | 0.305 (0.237) |
| | LF 3 | 0.814 (-0.001) | 0.409 (-0.006) | 0.054 (0.035) | 0.357 (0.231) |
| | LF 4 | 0.307 (-0.375) | 0.229 (0.032) | 0.085 (-0.517) | 0.042 (-0.600) |
| | LF 5 | 0.898 (0.000) | 0.311 (0.010) | 0.133 (0.081) | 0.236 (0.173) |
| | LF 6 | 0.677 (0.102) | 0.105 (-0.102) | 0.003 (-0.180) | 0.122 (-0.096) |

Table 12: NLP dataset raw LF performance

| Dataset | LF | Acc | F1 | $\Delta_{DP}$ | $\Delta_{EO}$ |
|---|---|---|---|---|---|
| CivilComments | LF 1 (sex LF) | 0.755 | 0.187 | 0.046 | 0.019 |
| | LF 2 (LGBTQ LF) | 0.877 | 0.073 | 0.001 | 0.017 |
| | LF 3 (religion LF) | 0.861 | 0.049 | 0.012 | 0.013 |
| | LF 4 (race LF) | 0.847 | 0.234 | 0.634 | 0.574 |
| | LF 5 (toxic LF) | 0.886 | 0.068 | 0.006 | 0.012 |
| | LF 6 (threat LF) | 0.862 | 0.102 | 0.055 | 0.054 |
| | LF 7 (insult LF) | 0.872 | 0.176 | 0.028 | 0.035 |
| HateXplain | LF 1 (sex LF) | 0.427 | 0.253 | 0.015 | 0.002 |
| | LF 2 (LGBTQ LF) | 0.405 | 0.077 | 0.047 | 0.041 |
| | LF 3 (religion LF) | 0.437 | 0.197 | 0.001 | 0.003 |
| | LF 4 (race LF) | 0.419 | 0.327 | 0.139 | 0.168 |
| | LF 5 (toxic LF) | 0.451 | 0.233 | 0.007 | 0.016 |
| | LF 6 (threat LF) | 0.415 | 0.097 | 0.000 | 0.004 |
| | LF 7 (insult LF) | 0.427 | 0.100 | 0.015 | 0.005 |
| | LF 8 (Detoxify - original) | 0.645 | 0.704 | 0.165 | 0.086 |
| | LF 9 (Detoxify - unbiased) | 0.625 | 0.668 | 0.150 | 0.078 |
| | LF 10 (Detoxify - multilingual) | 0.649 | 0.700 | 0.168 | 0.077 |
| | LF 11 (Detoxify - original-small) | 0.644 | 0.705 | 0.152 | 0.076 |
| | LF 12 (Detoxify - unbiased-small) | 0.643 | 0.699 | 0.186 | 0.113 |

Table 13: NLP dataset SBM (w/o OT) LF performance. $\Delta$s are obtained by comparison with raw LF performance.

| Dataset | LF | Acc ($\Delta$) | F1 ($\Delta$) | $\Delta_{DP}$ ($\Delta$) | $\Delta_{EO}$ ($\Delta$) |
|---|---|---|---|---|---|
| Civil Comments | LF 1 | 0.790 (0.035) | 0.325 (0.138) | 0.048 (0.002) | 0.041 (0.022) |
| | LF 2 | 0.896 (0.019) | 0.268 (0.195) | 0.001 (0.000) | 0.053 (0.036) |
| | LF 3 | 0.868 (0.007) | 0.155 (0.106) | 0.012 (0.000) | 0.043 (0.030) |
| | LF 4 | 0.858 (0.011) | 0.252 (0.018) | 0.114 (-0.520) | 0.160 (-0.414) |
| | LF 5 | 0.886 (0.000) | 0.137 (0.069) | 0.006 (0.000) | 0.003 (-0.009) |
| | LF 6 | 0.916 (0.054) | 0.482 (0.380) | 0.023 (-0.032) | 0.002 (-0.052) |
| | LF 7 | 0.918 (0.046) | 0.501 (0.325) | 0.022 (-0.006) | 0.012 (-0.023) |
| HateXplain | LF 1 | 0.483 (0.056) | 0.273 (0.020) | 0.015 (0.000) | 0.001 (-0.001) |
| | LF 2 | 0.473 (0.068) | 0.058 (-0.019) | 0.000 (-0.047) | 0.004 (-0.037) |
| | LF 3 | 0.460 (0.023) | 0.163 (-0.034) | 0.001 (0.000) | 0.004 (0.001) |
| | LF 4 | 0.481 (0.062) | 0.328 (0.001) | 0.044 (-0.095) | 0.039 (-0.129) |
| | LF 5 | 0.515 (0.064) | 0.267 (0.034) | 0.014 (0.007) | 0.021 (0.005) |
| | LF 6 | 0.471 (0.056) | 0.106 (0.009) | 0.000 (0.000) | 0.015 (0.011) |
| | LF 7 | 0.472 (0.045) | 0.088 (-0.012) | 0.012 (-0.003) | 0.014 (0.009) |
| | LF 8 | 0.831 (0.186) | 0.864 (0.160) | 0.006 (-0.159) | 0.000 (-0.086) |
| | LF 9 | 0.917 (0.292) | 0.928 (0.260) | 0.015 (-0.135) | 0.000 (-0.078) |
| | LF 10 | 0.864 (0.215) | 0.888 (0.188) | 0.009 (-0.159) | 0.000 (-0.077) |
| | LF 11 | 0.839 (0.195) | 0.870 (0.165) | 0.015 (-0.137) | 0.000 (-0.076) |
| | LF 12 | 0.837 (0.194) | 0.868 (0.169) | 0.014 (-0.172) | 0.000 (-0.113) |

Table 14: NLP dataset SBM (OT-L) LF performance. $\Delta$s are obtained by comparison with raw LF performance.

| Dataset | LF | Acc ($\Delta$) | F1 ($\Delta$) | $\Delta_{DP}$ ($\Delta$) | $\Delta_{EO}$ ($\Delta$) |
|---|---|---|---|---|---|
| Civil Comments | LF 1 | 0.791 (0.036) | 0.321 (0.134) | 0.003 (-0.043) | 0.022 (0.003) |
| | LF 2 | 0.898 (0.021) | 0.272 (0.199) | 0.001 (0.000) | 0.020 (0.003) |
| | LF 3 | 0.870 (0.009) | 0.156 (0.107) | 0.012 (0.000) | 0.041 (0.028) |
| | LF 4 | 0.860 (0.013) | 0.244 (0.010) | 0.017 (-0.617) | 0.017 (-0.557) |
| | LF 5 | 0.887 (0.001) | 0.139 (0.071) | 0.006 (0.000) | 0.027 (0.015) |
| | LF 6 | 0.917 (0.055) | 0.484 (0.382) | 0.012 (-0.043) | 0.026 (-0.028) |
| | LF 7 | 0.919 (0.047) | 0.501 (0.325) | 0.007 (-0.021) | 0.013 (-0.022) |
| HateXplain | LF 1 | 0.457 (0.030) | 0.279 (0.026) | 0.015 (0.000) | 0.001 (-0.001) |
| | LF 2 | 0.433 (0.028) | 0.062 (-0.015) | 0.002 (-0.045) | 0.006 (-0.035) |
| | LF 3 | 0.429 (-0.008) | 0.171 (-0.026) | 0.001 (0.000) | 0.001 (-0.002) |
| | LF 4 | 0.457 (0.038) | 0.323 (-0.004) | 0.011 (-0.128) | 0.009 (-0.159) |
| | LF 5 | 0.479 (0.028) | 0.263 (0.030) | 0.017 (0.010) | 0.029 (0.013) |
| | LF 6 | 0.436 (0.021) | 0.111 (0.014) | 0.000 (0.000) | 0.011 (0.007) |
| | LF 7 | 0.434 (0.007) | 0.091 (-0.009) | 0.012 (-0.003) | 0.017 (0.012) |
| | LF 8 | 0.845 (0.200) | 0.882 (0.178) | 0.790 (0.093) | 0.000 (-0.086) |
| | LF 9 | 0.923 (0.298) | 0.938 (0.270) | 0.883 (0.179) | 0.000 (-0.078) |
| | LF 10 | 0.875 (0.226) | 0.903 (0.203) | 0.823 (0.111) | 0.000 (-0.077) |
| | LF 11 | 0.848 (0.204) | 0.884 (0.179) | 0.792 (0.098) | 0.000 (-0.076) |
| | LF 12 | 0.845 (0.202) | 0.882 (0.183) | 0.789 (0.090) | 0.000 (-0.113) |

Table 15: NLP dataset SBM (OT-S) LF performance. $\Delta$s are obtained by comparison with raw LF performance.

| Dataset | LF | Acc ($\Delta$) | F1 ($\Delta$) | $\Delta_{DP}$ ($\Delta$) | $\Delta_{EO}$ ($\Delta$) |
|---|---|---|---|---|---|
| Civil Comments | LF 1 | 0.791 (0.036) | 0.320 (0.133) | 0.015 (-0.031) | 0.082 (0.063) |
| | LF 2 | 0.897 (0.020) | 0.271 (0.198) | 0.001 (0.000) | 0.029 (0.012) |
| | LF 3 | 0.870 (0.009) | 0.156 (0.107) | 0.012 (0.000) | 0.043 (0.030) |
| | LF 4 | 0.860 (0.013) | 0.245 (0.011) | 0.017 (-0.617) | 0.016 (-0.558) |
| | LF 5 | 0.887 (0.001) | 0.138 (0.070) | 0.006 (0.000) | 0.020 (0.008) |
| | LF 6 | 0.917 (0.055) | 0.482 (0.380) | 0.011 (-0.044) | 0.006 (-0.048) |
| | LF 7 | 0.919 (0.047) | 0.504 (0.328) | 0.019 (-0.009) | 0.039 (0.004) |
| HateXplain | LF 1 | 0.444 (0.017) | 0.277 (0.024) | 0.015 (0.000) | 0.002 (0.000) |
| | LF 2 | 0.417 (0.012) | 0.056 (-0.021) | 0.001 (-0.046) | 0.001 (-0.040) |
| | LF 3 | 0.418 (-0.019) | 0.172 (-0.025) | 0.001 (0.000) | 0.002 (-0.001) |
| | LF 4 | 0.448 (0.029) | 0.332 (0.005) | 0.032 (-0.107) | 0.045 (-0.123) |
| | LF 5 | 0.459 (0.008) | 0.256 (0.023) | 0.030 (0.023) | 0.041 (0.025) |
| | LF 6 | 0.422 (0.007) | 0.110 (0.013) | 0.000 (0.000) | 0.011 (0.007) |
| | LF 7 | 0.418 (-0.009) | 0.091 (-0.009) | 0.018 (0.003) | 0.025 (0.020) |
| | LF 8 | 0.851 (0.206) | 0.889 (0.185) | 0.056 (-0.109) | 0.000 (-0.086) |
| | LF 9 | 0.927 (0.302) | 0.942 (0.274) | 0.061 (-0.089) | 0.000 (-0.078) |
| | LF 10 | 0.884 (0.235) | 0.911 (0.211) | 0.052 (-0.116) | 0.000 (-0.077) |
| | LF 11 | 0.853 (0.209) | 0.890 (0.185) | 0.056 (-0.096) | 0.000 (-0.076) |
| | LF 12 | 0.847 (0.204) | 0.886 (0.187) | 0.063 (-0.123) | 0.000 (-0.113) |

Table 16: Vision dataset raw LF performance

| Dataset | LF | Acc | F1 | $\Delta_{DP}$ | $\Delta_{EO}$ |
|---|---|---|---|---|---|
| CelebA | LF 1 (ResNet-18 fine-tuned on gender classification dataset) | 0.798 | 0.794 | 0.328 | 0.284 |
| | LF 2 (ResNet-34 fine-tuned on FairFace dataset) | 0.890 | 0.901 | 0.314 | 0.105 |
| | LF 3 (ResNet-34 fine-tuned on UTKFace dataset) | 0.826 | 0.831 | 0.309 | 0.195 |
| | LF 4 (ResNet-34 fine-tuned on UTKFace dataset (White only)) | 0.825 | 0.832 | 0.277 | 0.131 |
| | LF 5 (ResNet-34 fine-tuned on UTKFace dataset (non-White only)) | 0.818 | 0.832 | 0.271 | 0.134 |
| | LF 6 (ResNet-34 fine-tuned on UTKFace dataset (age $\geq 50$ only) | 0.764 | 0.750 | 0.279 | 0.194 |
| | LF 7 (ResNet-34 fine-tuned on UTKFace dataset (age $< 50$ only)) | 0.830 | 0.845 | 0.299 | 0.175 |
| UTKFace | LF 1 (ResNet-18 fine-tuned on gender classification dataset) | 0.869 | 0.856 | 0.060 | 0.039 |
| | LF 2 (ResNet-34 fine-tuned on gender classification dataset) | 0.854 | 0.842 | 0.060 | 0.060 |
| | LF 3 (ResNet-34 fine-tuned on CelebA dataset) | 0.742 | 0.758 | 0.158 | 0.032 |
| | LF 4 (ResNet-34 fine-tuned on CelebA dataset (attractive only)) | 0.580 | 0.692 | 0.065 | 0.002 |
| | LF 5 (ResNet-34 fine-tuned on CelebA dataset (non-attractive only)) | 0.687 | 0.608 | 0.129 | 0.034 |
| | LF 6 (ResNet-34 fine-tuned on CelebA dataset (young only)) | 0.664 | 0.729 | 0.116 | 0.012 |
| | LF 7 (ResNet-34 fine-tuned on CelebA dataset (non-young only)) | 0.619 | 0.429 | 0.136 | 0.081 |
| | LF 8 (ResNet-34 fine-tuned on CelebA dataset (unfair sampling)) | 0.631 | 0.676 | 0.113 | 0.053 |

Table 17: Vision dataset SBM LF (w/o OT) performance. $\Delta$s are obtained by comparison with raw LF performance.

| Dataset | LF | Acc ($\Delta$) | F1 ($\Delta$) | $\Delta_{DP}$ ($\Delta$) | $\Delta_{EO}$ ($\Delta$) |
|---|---|---|---|---|---|
| CelebA | LF 1 | 0.847 (0.049) | 0.832 (0.038) | 0.328 (0.000) | 0.267 (-0.017) |
| | LF 2 | 0.890 (0.000) | 0.895 (-0.006) | 0.314 (0.000) | 0.101 (-0.004) |
| | LF 3 | 0.926 (0.100) | 0.923 (0.092) | 0.309 (0.000) | 0.097 (-0.098) |
| | LF 4 | 0.914 (0.089) | 0.912 (0.080) | 0.277 (0.000) | 0.027 (-0.104) |
| | LF 5 | 0.899 (0.081) | 0.900 (0.068) | 0.271 (0.000) | 0.030 (-0.104) |
| | LF 6 | 0.705 (-0.059) | 0.629 (-0.121) | 0.177 (-0.102) | 0.052 (-0.142) |
| | LF 7 | 0.913 (0.083) | 0.915 (0.070) | 0.299 (0.000) | 0.056 (-0.119) |
| UTKFace | LF 1 | 0.929 (0.060) | 0.924 (0.068) | 0.102 (0.042) | 0.011 (-0.028) |
| | LF 2 | 0.939 (0.085) | 0.935 (0.093) | 0.102 (0.042) | 0.007 (-0.053) |
| | LF 3 | 0.631 (-0.111) | 0.678 (-0.080) | 0.078 (-0.080) | 0.034 (0.002) |
| | LF 4 | 0.549 (-0.031) | 0.681 (-0.011) | 0.017 (-0.048) | 0.002 (0.000) |
| | LF 5 | 0.740 (0.053) | 0.679 (0.071) | 0.129 (0.000) | 0.040 (0.006) |
| | LF 6 | 0.694 (0.030) | 0.755 (0.026) | 0.116 (0.000) | 0.037 (0.025) |
| | LF 7 | 0.694 (0.075) | 0.541 (0.112) | 0.061 (-0.075) | 0.033 (-0.048) |
| | LF 8 | 0.591 (-0.040) | 0.654 (-0.022) | 0.071 (-0.042) | 0.054 (0.001) |

Table 18: Vision dataset SBM (OT-L) LF performance. $\Delta$s are obtained by comparison with raw LF performance.

| Dataset | LF | Acc ($\Delta$) | F1 ($\Delta$) | $\Delta_{DP}$ ($\Delta$) | $\Delta_{EO}$ ($\Delta$) |
|---|---|---|---|---|---|
| CelebA | LF 1 | 0.847 (0.049) | 0.832 (0.038) | 0.328 (0.000) | 0.268 (-0.016) |
| | LF 2 | 0.890 (0.000) | 0.894 (-0.007) | 0.314 (0.000) | 0.102 (-0.003) |
| | LF 3 | 0.926 (0.100) | 0.922 (0.091) | 0.309 (0.000) | 0.098 (-0.097) |
| | LF 4 | 0.915 (0.090) | 0.913 (0.081) | 0.277 (0.000) | 0.029 (-0.102) |
| | LF 5 | 0.898 (0.080) | 0.900 (0.068) | 0.271 (0.000) | 0.031 (-0.103) |
| | LF 6 | 0.648 (-0.116) | 0.498 (-0.252) | 0.059 (-0.220) | 0.217 (0.023) |
| | LF 7 | 0.914 (0.084) | 0.916 (0.071) | 0.299 (0.000) | 0.058 (-0.117) |
| UTKFace | LF 1 | 0.931 (0.062) | 0.925 (0.069) | 0.026 (-0.034) | 0.012 (-0.027) |
| | LF 2 | 0.945 (0.091) | 0.940 (0.098) | 0.017 (-0.043) | 0.006 (-0.054) |
| | LF 3 | 0.599 (-0.143) | 0.667 (-0.091) | 0.004 (-0.154) | 0.001 (-0.031) |
| | LF 4 | 0.523 (-0.057) | 0.667 (-0.025) | 0.008 (-0.057) | 0.002 (0.000) |
| | LF 5 | 0.738 (0.051) | 0.674 (0.066) | 0.129 (0.000) | 0.029 (-0.005) |
| | LF 6 | 0.691 (0.027) | 0.752 (0.023) | 0.116 (0.000) | 0.037 (0.025) |
| | LF 7 | 0.690 (0.071) | 0.525 (0.096) | 0.000 (-0.136) | 0.016 (-0.065) |
| | LF 8 | 0.572 (-0.059) | 0.655 (-0.021) | 0.001 (-0.112) | 0.014 (-0.039) |

Table 19: Vision dataset SBM (OT-S) LF performance. $\Delta$s are obtained by comparison with raw LF performance.

| Dataset | LF | Acc ($\Delta$) | F1 ($\Delta$) | $\Delta_{DP}$ ($\Delta$) | $\Delta_{EO}$ ($\Delta$) |
|---|---|---|---|---|---|
| CelebA | LF 1 | 0.850 (0.052) | 0.835 (0.041) | 0.328 (0.000) | 0.266 (-0.018) |
| | LF 2 | 0.893 (0.003) | 0.897 (-0.004) | 0.314 (0.000) | 0.101 (-0.004) |
| | LF 3 | 0.923 (0.097) | 0.920 (0.089) | 0.309 (0.000) | 0.098 (-0.097) |
| | LF 4 | 0.913 (0.088) | 0.910 (0.078) | 0.277 (0.000) | 0.029 (-0.102) |
| | LF 5 | 0.901 (0.083) | 0.903 (0.071) | 0.271 (0.000) | 0.031 (-0.103) |
| | LF 6 | 0.618 (-0.146) | 0.430 (-0.320) | 0.016 (-0.263) | 0.282 (0.088) |
| | LF 7 | 0.911 (0.081) | 0.913 (0.068) | 0.299 (0.000) | 0.059 (-0.116) |
| UTKFace | LF 1 | 0.931 (0.062) | 0.926 (0.070) | 0.010 (-0.050) | 0.001 (-0.038) |
| | LF 2 | 0.949 (0.095) | 0.945 (0.103) | 0.015 (-0.045) | 0.001 (-0.059) |
| | LF 3 | 0.577 (-0.165) | 0.652 (-0.106) | 0.003 (-0.155) | 0.013 (-0.019) |
| | LF 4 | 0.517 (-0.063) | 0.664 (-0.028) | 0.006 (-0.059) | 0.007 (0.005) |
| | LF 5 | 0.730 (0.043) | 0.669 (0.061) | 0.129 (0.000) | 0.018 (-0.016) |
| | LF 6 | 0.694 (0.030) | 0.756 (0.027) | 0.116 (0.000) | 0.036 (0.024) |
| | LF 7 | 0.683 (0.064) | 0.523 (0.094) | 0.010 (-0.126) | 0.007 (-0.074) |
| | LF 8 | 0.562 (-0.069) | 0.652 (-0.024) | 0.009 (-0.104) | 0.012 (-0.041) |

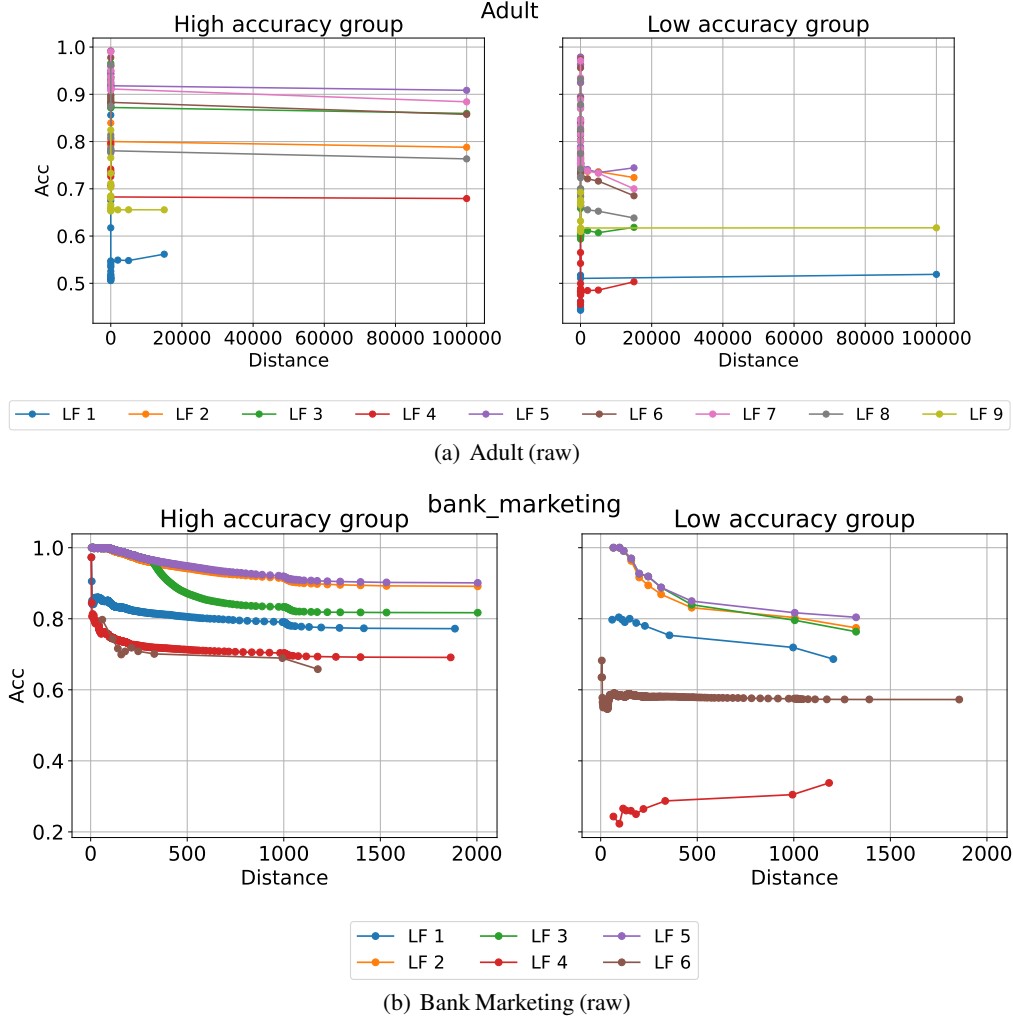

(a) Adult (raw)

(b) Bank Marketing (raw)

Figure 6: Identification of high accuracy regimes for tabular datasets.

## E.4 Identification of centers

While our method improves performance by matching one group distribution and attempting to make these uniform, it does not imply accuracy improvement. A presumption of our method is that the group with high (estimated) accuracy possesses the high accuracy regime and our method can transport data points to this high accuracy regime while keeping their structure, which results in accuracy improvements. To empirically support this hypothesis, we used the following procedure and visualized the results in Figure 6, 8, 9.

1. Find the best accuracy center by evaluating the accuracy of the nearest 10% of data points for each center candidate point.
2. Expand 2% percent of data points closest to the center each time, compute their accumulated average accuracy (y-axis) and the farthest distance (x-axis) from the neighborhood
3. Find the group with better accuracy group and visualize their accumulated average accuracy (y-axis) and the farthest distance (x-axis) in each group.

We are able to obtain two insights from the visualization. First, the high accuracy regime typically exists in the group with the high estimated accuracy, which supports our hypothesis. Thus accuracy improvement by optimal transport can be justified. Secondly, the groups actually show the distributional difference in the input space $\mathcal{X}$. Given center points, lines in the high accuracy group start with a smaller distance to the center than the low accuracy group.

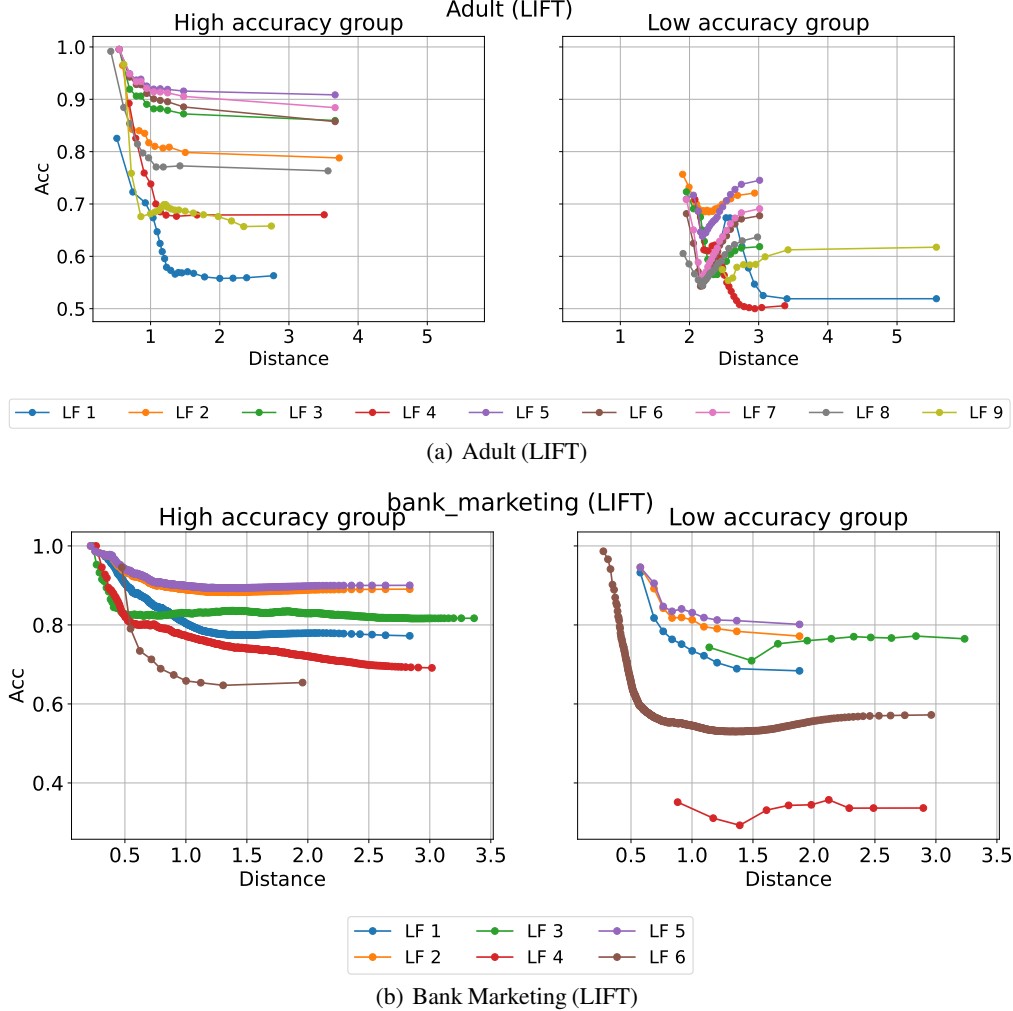

(a) Adult (LIFT)

(b) Bank Marketing (LIFT)

Figure 7: Identification of high accuracy regimes for tabular datasets (LIFT).

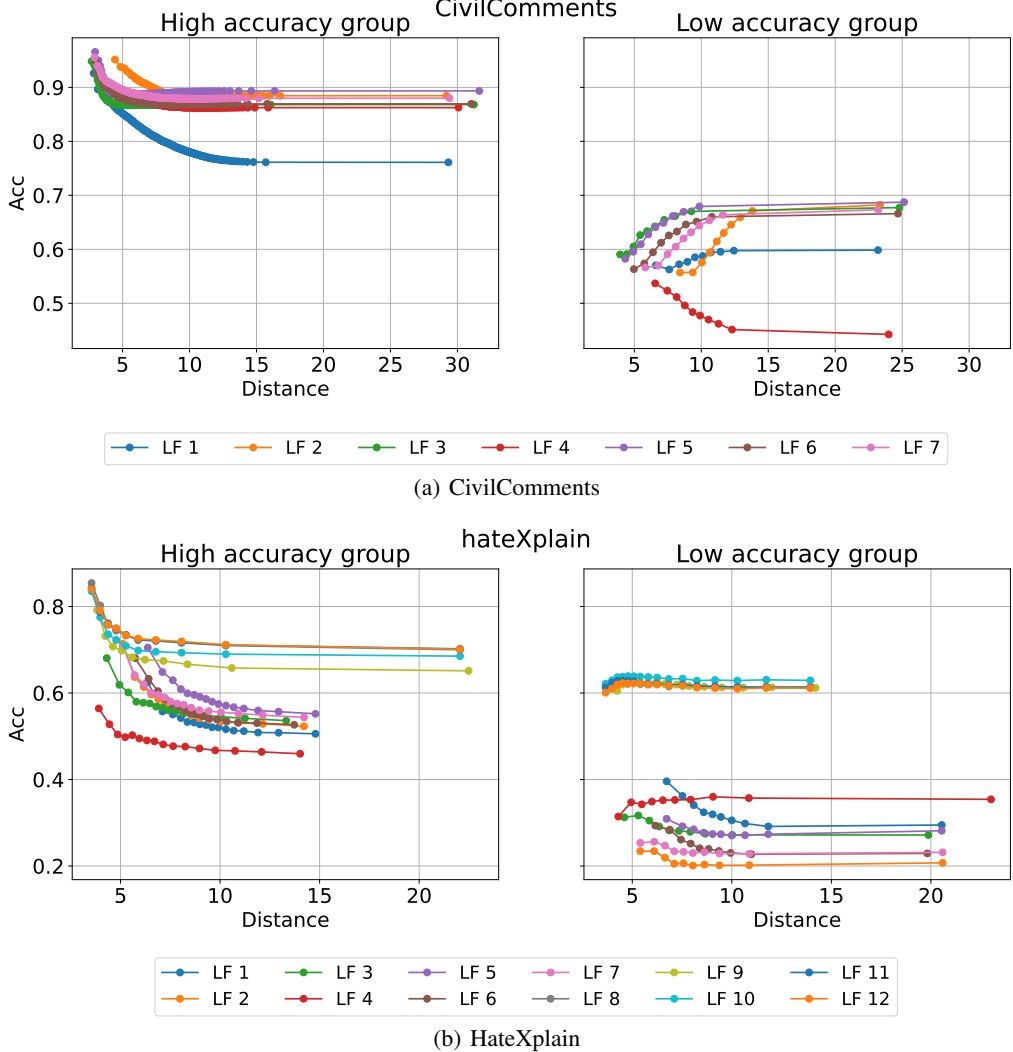

Figure 8: Identification of high accuracy regimes for NLP datasets.

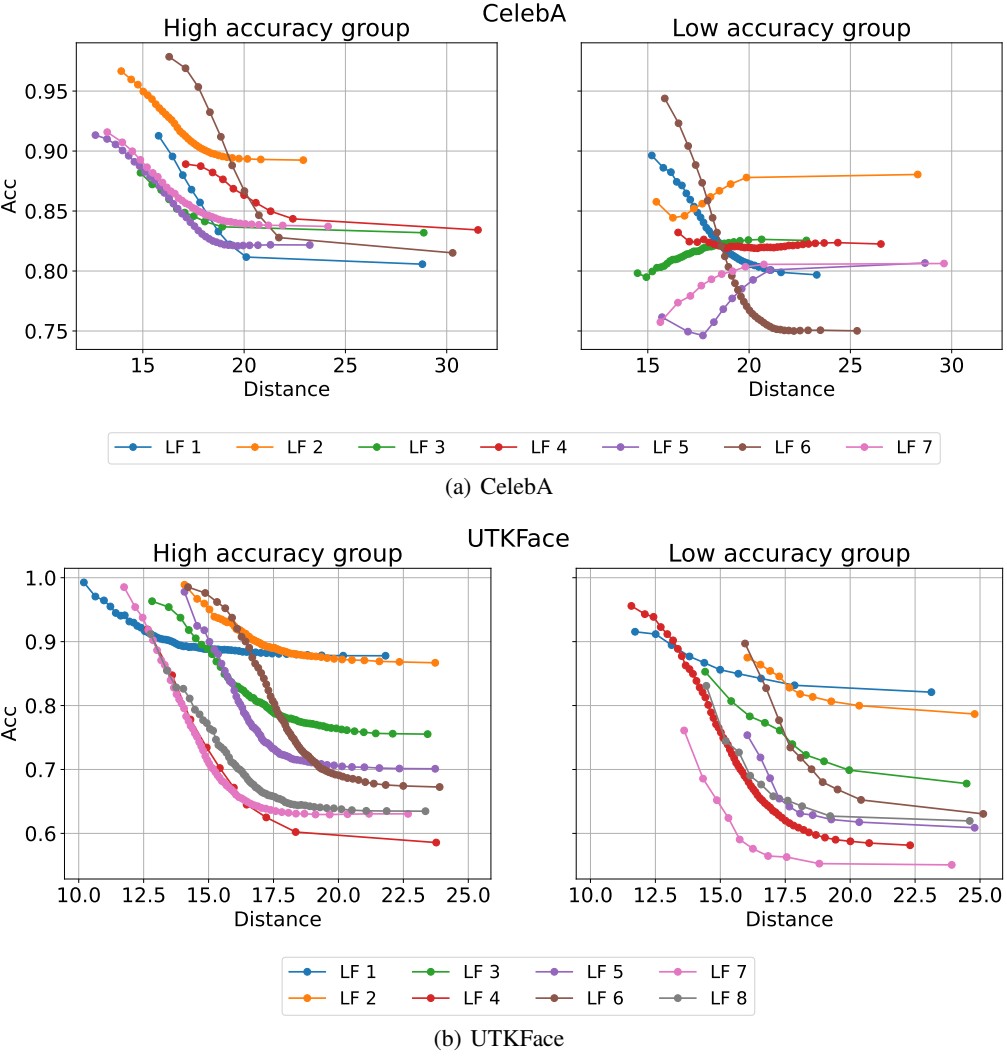

Figure 9: Identification of high accuracy regimes for vision datasets.

### E.5 Compatibility with other fair ML methods

One advantage of our method is that we can use other successful fair ML methods in a supervised learning setting on top of SBM, since our method works in weak label sources while traditional fair ML methods work in the preprocessing/training/postprocessing steps, which are independent of the label model. To make this point, we tried traditional fair ML methods from fairlearn [BDE$^+$20] with each of WS settings. We used CorrelationRemover, ExponentiatedGradient [ABD$^+$18], ThresholdOptimizer [HPS16] with the demographic parity (DP), equal opportunity (EO) as parity criteria, and accuracy as the performance criteria. The results are reported in Table 20 - 27. As expected, combining with other methods yields an accuracy-fairness tradeoff given weak label sources. Typically, SBM yields additional gains upon traditional fair ML methods. One another observation is that fair ML methods to modify equal opportunity typically fail to achieve the reduction of $\Delta_{EO}$. This can be interpreted as the result of the noise in the training set labels.

Table 20: SBM combined with other fair ML methods in Adult dataset

|  | Fair ML method | Acc | F1 | $\Delta_{DP}$ | $\Delta_{EO}$ |
|---|---|---|---|---|---|
| WS (Baseline) | N/A | 0.717 | 0.587 | 0.475 | 0.325 |
|  | correlation remover | 0.716 | 0.587 | 0.446 | 0.287 |
|  | optimal threshold (DP) | 0.578 | 0.499 | **0.002** | 0.076 |
|  | optimal threshold (EO) | 0.721 | 0.563 | 0.404 | 0.217 |
|  | exponentiated gradient (DP gap = 0) | 0.582 | 0.502 | **0.002** | 0.066 |
|  | exponentiated gradient (EO gap = 0) | 0.715 | 0.585 | 0.445 | 0.284 |
| SBM (w/o OT) | N/A | 0.720 | 0.592 | 0.439 | 0.273 |
|  | correlation remover | 0.717 | 0.586 | 0.437 | 0.264 |
|  | optimal threshold (DP) | 0.591 | 0.507 | 0.003 | 0.059 |
|  | optimal threshold (EO) | 0.722 | 0.571 | 0.387 | 0.189 |
|  | exponentiated gradient (DP gap = 0) | 0.693 | 0.525 | 0.014 | 0.052 |
|  | exponentiated gradient (EO gap = 0) | 0.722 | 0.586 | 0.404 | 0.233 |
| SBM (OT-L) | N/A | 0.560 | 0.472 | 0.893 | 0.980 |
|  | correlation remover | 0.460 | 0.443 | 0.084 | **0.005** |
|  | optimal threshold (DP) | 0.324 | 0.404 | 0.006 | 0.089 |
|  | optimal threshold (EO) | 0.300 | 0.399 | 0.103 | 0.015 |
|  | exponentiated gradient (DP gap = 0) | 0.345 | 0.414 | **0.002** | 0.016 |
|  | exponentiated gradient (EO gap = 0) | 0.558 | 0.479 | 0.861 | 0.792 |
| SBM (OT-S) | N/A | 0.722 | 0.590 | 0.429 | 0.261 |
|  | correlation remover | **0.729** | **0.595** | 0.408 | 0.249 |
|  | optimal threshold (DP) | 0.596 | 0.507 | 0.003 | 0.050 |
|  | optimal threshold (EO) | 0.723 | 0.571 | 0.382 | 0.184 |
|  | exponentiated gradient (DP gap = 0) | 0.687 | 0.527 | 0.011 | 0.045 |
|  | exponentiated gradient (EO gap = 0) | 0.728 | 0.587 | 0.390 | 0.218 |

Table 21: SBM combined with other fair ML methods in Adult dataset (LIFT)

| | Fair ML method | Acc | F1 | $\Delta_{DP}$ | $\Delta_{EO}$ |
|---|---|---|---|---|---|
| **WS (Baseline)** | N/A | 0.711 | 0.584 | 0.449 | 0.290 |
| | correlation remover | 0.716 | **0.587** | 0.446 | 0.287 |
| | optimal threshold (DP) | 0.578 | 0.499 | 0.002 | 0.076 |
| | optimal threshold (EO) | 0.721 | 0.563 | 0.404 | 0.217 |
| | exponentiated gradient (DP gap = 0) | 0.582 | 0.502 | 0.002 | 0.066 |
| | exponentiated gradient (EO gap = 0) | 0.715 | 0.585 | 0.445 | 0.284 |
| **SBM (w/o OT)** | N/A | 0.704 | 0.366 | 0.032 | 0.192 |
| | correlation remover | 0.686 | 0.351 | 0.006 | 0.155 |
| | optimal threshold (DP) | 0.707 | 0.363 | 0.007 | 0.133 |
| | optimal threshold (EO) | 0.713 | 0.362 | 0.022 | 0.079 |
| | exponentiated gradient (DP gap = 0) | 0.682 | 0.350 | 0.011 | 0.163 |
| | exponentiated gradient (EO gap = 0) | 0.701 | 0.369 | 0.019 | 0.134 |
| **SBM (OT-L)** | N/A | 0.700 | 0.520 | 0.015 | 0.138 |
| | correlation remover | 0.686 | 0.504 | 0.011 | 0.105 |
| | optimal threshold (DP) | 0.701 | 0.520 | 0.008 | 0.124 |
| | optimal threshold (EO) | 0.712 | 0.521 | 0.060 | **0.025** |
| | exponentiated gradient (DP gap = 0) | 0.673 | 0.504 | 0.005 | 0.071 |
| | exponentiated gradient (EO gap = 0) | 0.691 | 0.516 | 0.058 | 0.035 |
| **SBM (OT-S)** | N/A | 0.782 | 0.448 | **0.000** | 0.178 |
| | correlation remover | 0.772 | 0.435 | 0.002 | 0.180 |
| | optimal threshold (DP) | 0.782 | 0.447 | 0.001 | 0.176 |
| | optimal threshold (EO) | **0.790** | 0.427 | 0.087 | 0.104 |
| | exponentiated gradient (DP gap = 0) | 0.784 | 0.452 | **0.000** | 0.171 |
| | exponentiated gradient (EO gap = 0) | 0.747 | 0.380 | 0.107 | 0.049 |

Table 22: SBM combined with other fair ML methods in Bank Marketing dataset

| | Fair ML method | Acc | F1 | $\Delta_{DP}$ | $\Delta_{EO}$ |
|---|---|---|---|---|---|
| **WS (Baseline)** | N/A | 0.674 | 0.258 | 0.543 | 0.450 |
| | correlation remover | 0.890 | 0.057 | 0.002 | **0.006** |
| | optimal threshold (DP) | 0.890 | 0.058 | 0.002 | 0.007 |
| | optimal threshold (EO) | 0.890 | 0.066 | 0.030 | 0.040 |
| | exponentiated gradient (DP gap = 0) | 0.889 | 0.039 | **0.000** | 0.009 |
| | exponentiated gradient (EO gap = 0) | 0.890 | 0.070 | 0.033 | 0.051 |
| **SBM (w/o OT)** | N/A | 0.876 | **0.550** | 0.106 | 0.064 |
| | correlation remover | 0.874 | 0.547 | 0.064 | 0.095 |
| | optimal threshold (DP) | 0.876 | 0.547 | 0.031 | 0.208 |
| | optimal threshold (EO) | 0.876 | 0.548 | 0.053 | 0.171 |
| | exponentiated gradient (DP gap = 0) | 0.877 | 0.525 | 0.037 | 0.182 |
| | exponentiated gradient (EO gap = 0) | 0.872 | 0.531 | 0.066 | 0.124 |
| **SBM (OT-L)** | N/A | 0.892 | 0.304 | 0.095 | 0.124 |
| | correlation remover | 0.890 | 0.290 | 0.011 | 0.111 |
| | optimal threshold (DP) | 0.891 | 0.280 | 0.008 | 0.163 |
| | optimal threshold (EO) | 0.881 | 0.313 | 0.841 | 0.678 |
| | exponentiated gradient (DP gap = 0) | 0.891 | 0.263 | 0.003 | 0.106 |
| | exponentiated gradient (EO gap = 0) | **0.895** | 0.296 | 0.097 | 0.136 |
| **SBM (OT-S)** | N/A | 0.847 | 0.515 | 0.122 | 0.080 |
| | correlation remover | 0.847 | 0.512 | 0.072 | 0.122 |
| | optimal threshold (DP) | 0.846 | 0.511 | 0.043 | 0.236 |
| | optimal threshold (EO) | 0.847 | 0.515 | 0.113 | 0.104 |
| | exponentiated gradient (DP gap = 0) | 0.843 | 0.487 | 0.052 | 0.143 |
| | exponentiated gradient (EO gap = 0) | 0.848 | 0.512 | 0.114 | 0.088 |

Table 23: SBM combined with other fair ML methods in Bank Marketing dataset (LIFT)

|  | Fair ML method | Acc | F1 | $\Delta_{DP}$ | $\Delta_{EO}$ |
|---|---|---|---|---|---|
| WS (Baseline) | N/A | 0.674 | 0.258 | 0.543 | 0.450 |
|  | correlation remover | 0.890 | 0.057 | 0.002 | 0.006 |
|  | optimal threshold (DP) | 0.890 | 0.058 | 0.002 | 0.007 |
|  | optimal threshold (EO) | 0.890 | 0.066 | 0.030 | 0.040 |
|  | exponentiated gradient (DP gap = 0) | 0.889 | 0.039 | **0.000** | 0.009 |
|  | exponentiated gradient (EO gap = 0) | 0.890 | 0.070 | 0.033 | 0.051 |
| SBM (w/o OT) | N/A | 0.698 | 0.255 | 0.088 | 0.137 |
|  | correlation remover | 0.836 | **0.358** | 0.025 | 0.114 |
|  | optimal threshold (DP) | 0.699 | 0.252 | 0.002 | 0.019 |
|  | optimal threshold (EO) | 0.698 | 0.253 | 0.006 | 0.028 |
|  | exponentiated gradient (DP gap = 0) | 0.687 | 0.262 | 0.014 | 0.053 |
|  | exponentiated gradient (EO gap = 0) | 0.654 | 0.226 | 0.096 | 0.107 |
| SBM (OT-L) | N/A | 0.892 | 0.305 | 0.104 | 0.121 |
|  | correlation remover | 0.891 | 0.304 | 0.079 | **0.000** |
|  | optimal threshold (DP) | 0.891 | 0.289 | 0.016 | 0.094 |
|  | optimal threshold (EO) | 0.892 | 0.305 | 0.103 | 0.121 |
|  | exponentiated gradient (DP gap = 0) | **0.893** | 0.265 | 0.001 | 0.093 |
|  | exponentiated gradient (EO gap = 0) | 0.892 | 0.305 | 0.100 | 0.109 |
| SBM (OT-S) | N/A | 0.698 | 0.080 | 0.109 | 0.072 |
|  | correlation remover | 0.699 | 0.081 | 0.230 | 0.106 |
|  | optimal threshold (DP) | 0.697 | 0.083 | 0.028 | 0.011 |
|  | optimal threshold (EO) | 0.695 | 0.089 | 0.174 | 0.205 |
|  | exponentiated gradient (DP gap = 0) | 0.681 | 0.113 | 0.032 | 0.063 |
|  | exponentiated gradient (EO gap = 0) | 0.691 | 0.124 | 0.041 | 0.036 |

Table 24: SBM combined with other fair ML methods in CivilComments dataset

|  | Fair ML method | Acc | F1 | $\Delta_{DP}$ | $\Delta_{EO}$ |
|---|---|---|---|---|---|
| WS (Baseline) | N/A | 0.854 | **0.223** | 0.560 | 0.546 |
|  | correlation remover | 0.886 | 0.000 | 0.000 | 0.000 |
|  | optimal threshold (DP) | 0.886 | 0.000 | 0.000 | 0.000 |
|  | optimal threshold (EO) | 0.886 | 0.000 | 0.000 | 0.000 |
|  | exponentiated gradient (DP gap = 0) | 0.886 | 0.000 | 0.000 | 0.000 |
|  | exponentiated gradient (EO gap = 0) | 0.886 | 0.000 | 0.000 | 0.000 |
| SBM (w/o OT) | N/A | 0.879 | 0.068 | 0.048 | 0.047 |
|  | correlation remover | 0.878 | 0.062 | 0.010 | 0.030 |
|  | optimal threshold (DP) | 0.880 | 0.054 | 0.001 | 0.015 |
|  | optimal threshold (EO) | 0.880 | 0.061 | 0.019 | 0.010 |
|  | exponentiated gradient (DP gap = 0) | 0.881 | 0.046 | 0.002 | 0.008 |
|  | exponentiated gradient (EO gap = 0) | 0.880 | 0.059 | 0.018 | 0.010 |
| SBM (OT-L) | N/A | 0.880 | 0.070 | 0.042 | 0.039 |
|  | correlation remover | 0.879 | 0.056 | 0.011 | 0.028 |
|  | optimal threshold (DP) | 0.880 | 0.060 | 0.001 | 0.017 |
|  | optimal threshold (EO) | 0.880 | 0.063 | 0.013 | 0.002 |
|  | exponentiated gradient (DP gap = 0) | **0.882** | 0.043 | 0.002 | 0.008 |
|  | exponentiated gradient (EO gap = 0) | **0.882** | 0.039 | 0.006 | 0.001 |
| SBM (OT-S) | N/A | **0.882** | 0.047 | 0.028 | 0.026 |
|  | correlation remover | 0.879 | 0.057 | 0.011 | 0.029 |
|  | optimal threshold (DP) | **0.882** | 0.040 | **0.000** | 0.011 |
|  | optimal threshold (EO) | **0.882** | 0.042 | 0.010 | **0.000** |
|  | exponentiated gradient (DP gap = 0) | 0.881 | 0.045 | 0.001 | 0.008 |
|  | exponentiated gradient (EO gap = 0) | 0.880 | 0.056 | 0.016 | 0.005 |

Table 25: SBM combined with other fair ML methods in HateXplain dataset

| | Fair ML method | Acc | F1 | $\Delta_{DP}$ | $\Delta_{EO}$ |
|---|---|---|---|---|---|
| WS (Baseline) | N/A | 0.584 | 0.590 | 0.171 | 0.133 |
| | correlation remover | 0.555 | 0.557 | 0.007 | 0.031 |
| | optimal threshold (DP) | 0.539 | 0.515 | 0.005 | 0.047 |
| | optimal threshold (EO) | 0.573 | 0.573 | 0.129 | 0.090 |
| | exponentiated gradient (DP gap = 0) | 0.562 | 0.561 | 0.006 | 0.055 |
| | exponentiated gradient (EO gap = 0) | 0.579 | 0.586 | 0.130 | 0.093 |
| SBM (w/o OT) | N/A | 0.592 | 0.637 | 0.159 | 0.138 |
| | correlation remover | 0.563 | 0.616 | 0.033 | 0.053 |
| | optimal threshold (DP) | 0.586 | 0.660 | 0.013 | 0.006 |
| | optimal threshold (EO) | 0.538 | 0.561 | 0.034 | 0.074 |
| | exponentiated gradient (DP gap = 0) | 0.581 | 0.638 | 0.039 | 0.013 |
| | exponentiated gradient (EO gap = 0) | 0.580 | 0.630 | 0.047 | 0.095 |
| SBM (OT-L) | N/A | 0.606 | 0.670 | 0.120 | 0.101 |
| | correlation remover | 0.587 | 0.657 | 0.057 | 0.087 |
| | optimal threshold (DP) | 0.600 | 0.683 | 0.010 | **0.004** |
| | optimal threshold (EO) | 0.563 | 0.615 | 0.039 | 0.071 |
| | exponentiated gradient (DP gap = 0) | 0.600 | 0.673 | 0.029 | 0.011 |
| | exponentiated gradient (EO gap = 0) | 0.593 | 0.669 | 0.044 | 0.087 |
| SBM (OT-S) | N/A | **0.612** | 0.687 | 0.072 | 0.037 |
| | correlation remover | 0.587 | 0.668 | 0.073 | 0.105 |
| | optimal threshold (DP) | 0.607 | 0.694 | **0.002** | 0.031 |
| | optimal threshold (EO) | 0.572 | **0.696** | 0.201 | 0.182 |
| | exponentiated gradient (DP gap = 0) | 0.598 | 0.683 | 0.005 | 0.020 |
| | exponentiated gradient (EO gap = 0) | 0.585 | 0.672 | 0.070 | 0.093 |

Table 26: SBM combined with other fair ML methods in CelebA dataset

| | Fair ML method | Acc | F1 | $\Delta_{DP}$ | $\Delta_{EO}$ |
|---|---|---|---|---|---|
| WS (Baseline) | N/A | 0.866 | 0.879 | 0.308 | 0.193 |
| | correlation remover | 0.845 | 0.862 | 0.099 | 0.066 |
| | optimal threshold (DP) | 0.816 | 0.845 | 0.009 | 0.035 |
| | optimal threshold (EO) | 0.789 | 0.793 | 0.196 | 0.033 |
| | exponentiated gradient (DP gap = 0) | 0.781 | 0.814 | 0.008 | **0.006** |
| | exponentiated gradient (EO gap = 0) | 0.838 | 0.854 | 0.205 | 0.025 |
| SBM (w/o OT) | N/A | 0.870 | 0.883 | 0.309 | 0.192 |
| | correlation remover | 0.849 | 0.865 | 0.095 | 0.066 |
| | optimal threshold (DP) | 0.819 | 0.848 | 0.009 | 0.038 |
| | optimal threshold (EO) | 0.792 | 0.798 | 0.194 | 0.030 |
| | exponentiated gradient (DP gap = 0) | 0.783 | 0.818 | **0.006** | 0.007 |
| | exponentiated gradient (EO gap = 0) | 0.841 | 0.857 | 0.206 | 0.029 |
| SBM (OT-L) | N/A | 0.870 | 0.883 | 0.306 | 0.185 |
| | correlation remover | 0.849 | 0.866 | 0.096 | 0.066 |
| | optimal threshold (DP) | 0.819 | 0.848 | 0.010 | 0.034 |
| | optimal threshold (EO) | 0.792 | 0.798 | 0.193 | 0.023 |
| | exponentiated gradient (DP gap = 0) | 0.783 | 0.818 | 0.007 | 0.008 |
| | exponentiated gradient (EO gap = 0) | 0.841 | 0.857 | 0.203 | 0.026 |
| SBM (OT-S) | N/A | **0.872** | **0.885** | 0.306 | 0.184 |
| | correlation remover | 0.851 | 0.867 | 0.097 | 0.062 |
| | optimal threshold (DP) | 0.821 | 0.850 | 0.010 | 0.035 |
| | optimal threshold (EO) | 0.795 | 0.801 | 0.193 | 0.023 |
| | exponentiated gradient (DP gap = 0) | 0.784 | 0.819 | 0.008 | 0.010 |
| | exponentiated gradient (EO gap = 0) | 0.840 | 0.857 | 0.200 | 0.022 |

Table 27: SBM combined with other fair ML methods in UTKFace dataset

| | Fair ML method | Acc | F1 | $\Delta_{DP}$ | $\Delta_{EO}$ |
|---|---|---|---|---|---|
| WS (Baseline) | N/A | 0.791 | 0.791 | 0.172 | 0.073 |
| | correlation remover | 0.787 | 0.786 | 0.034 | 0.051 |
| | optimal threshold (DP) | 0.774 | 0.767 | 0.040 | 0.215 |
| | optimal threshold (EO) | 0.788 | 0.786 | 0.114 | **0.005** |
| | exponentiated gradient (DP gap = 0) | 0.769 | 0.762 | 0.029 | 0.078 |
| | exponentiated gradient (EO gap = 0) | 0.792 | 0.790 | 0.126 | 0.026 |
| SBM (w/o OT) | N/A | 0.797 | 0.790 | 0.164 | 0.077 |
| | correlation remover | 0.791 | 0.793 | 0.006 | 0.091 |
| | optimal threshold (DP) | 0.764 | 0.791 | 0.033 | 0.202 |
| | optimal threshold (EO) | 0.789 | 0.791 | 0.165 | 0.077 |
| | exponentiated gradient (DP gap = 0) | 0.760 | 0.791 | 0.024 | 0.069 |
| | exponentiated gradient (EO gap = 0) | 0.791 | 0.792 | 0.158 | 0.076 |
| SBM (OT-L) | N/A | 0.800 | 0.793 | 0.135 | 0.043 |
| | correlation remover | 0.799 | 0.791 | **0.004** | 0.098 |
| | optimal threshold (DP) | 0.785 | 0.772 | 0.034 | 0.195 |
| | optimal threshold (EO) | 0.800 | 0.793 | 0.128 | 0.038 |
| | exponentiated gradient (DP gap = 0) | 0.779 | 0.764 | 0.024 | 0.069 |
| | exponentiated gradient (EO gap = 0) | 0.797 | 0.789 | 0.123 | 0.030 |
| SBM (OT-S) | N/A | **0.804** | 0.798 | 0.130 | 0.041 |
| | correlation remover | 0.799 | 0.794 | 0.012 | 0.087 |
| | optimal threshold (DP) | 0.789 | 0.777 | 0.036 | 0.195 |
| | optimal threshold (EO) | 0.799 | 0.794 | 0.168 | 0.046 |
| | exponentiated gradient (DP gap = 0) | 0.776 | 0.764 | 0.023 | 0.068 |
| | exponentiated gradient (EO gap = 0) | 0.801 | **0.796** | 0.141 | 0.044 |

### E.6 Additional synthetic experiment result on the number of LFs

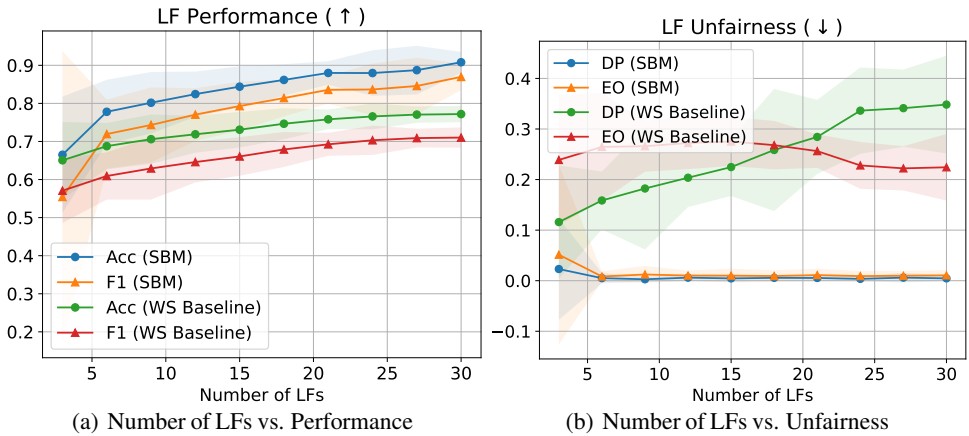

(a) Number of LFs vs. Performance      (b) Number of LFs vs. Unfairness

Figure 10: Synthetic experiment on number of LFs vs. performance and fairness

**Claim Investigated** One seemingly possible argument is that diversifying LFs can naturally resolve unfairness, weakening the necessity of our method. We hypothesized that simplying increasing the number of LFs does not improve fairness, while our method can further improve fairness as the number of LFs increases. To show this, we generate unfair synthetic data and increase the number of LFs and see if our method can remedy LF fairness and improve LF performance when LFs are diversified.

**Setup and Procedure** We generated a synthetic dataset ($n = 10000$) with the similar procedure as the previous synthetic experiment. Input samples in $\mathbb{R}^2$ are taken from $\sim \mathcal{N}(0, I)$, and true labels $Y$ are generated by $Y = 1[X[:, 0] \geq 0]$. Group transformation $g_b(x) : x + b$ is applied to a random half of samples, where $b \sim \mathcal{U}([10, 50]^2)$. We varied LFs by randomly sampling parameters based on our label model. Specifically, parameters for each LFs are sampled from $\theta_j \sim \mathcal{U}(0.1, 3)$, $x^{center_j} \sim \mathcal{U}([-5, 5]^2)$. We varied the number of LFs from 3 to 30. We repeat experiments 10 times with different seeds and provide means and confidence intervals.

**Results** The result is reported in Figure 10. The result shows that simply diversifying label sources does not improve fairness—though performance can be improved by adding more LFs. On the other hand, SBM resolves unfairness and yields better performance by mitigating source bias.

### E.7 Large language model experiment

Recently, large language models (LMs) pre-trained with massive datasets have shown excellent performance in many NLP tasks, even without fine-tuning. As a proof of concept that our method can be used for LMs, we conducted an additional experiment that applies our method to NLP tasks with LMs. [ANC⁺22] introduced AMA, which is a prompting strategy that provides multiple prompts with varying instructions, and combines their outputs to obtain a final answer using weak supervision. In this setup, distinct prompts can be seen as label sources. Based on their setup, we craft group annotations that identify a low accuracy regime by Barlow slicing [SNS⁺21], which is a tree-based data slicing method. After applying data slicing for each prompt, we apply SBM and use the AMA pipeline. The result is shown in Table 28. SBM with Barlow data slicing yields modest performance gains over AMA. This result suggests that it may be possible to use our techniques to gain improvements even in generic zero-shot language model prediction.

Table 28: SBM combined with AMA [ANC⁺22]. Performance metric is accuracy (%).

| Methods | RTE | WSC | WIC |
|---|---|---|---|
| AMA [ANC⁺22] | 75.1 | 77.9 | 61.3 |
| AMA (Reproduced) | 75.1 | 76 | 60.8 |
| SBM (w/o OT) | **75.5** | 71.2 | 61.1 |
| SBM (OT-L) | 75.1 | 70.2 | **61.6** |
| SBM (OT-S) | 74.7 | **78.8** | 61.1 |

