# OpenReview forum: "Mitigating Source Bias for Fairer Weak Supervision"
_NeurIPS.cc/2023/Conference — NeurIPS 2023 poster_

### Official Review · Reviewer_6e6W · 2023-06-12

**Soundness:** 4 excellent
**Presentation:** 3 good
**Contribution:** 3 good
**Rating:** 7
**Confidence:** 4

**Summary:**

This work found that unfair LFs in programmatic weak supervision could introduce bias to the resultant training labels, and proposed to address the bias via source bias mitigation and provided theoretical guarantee. Experimental results show that the effectiveness of the approaches in both synthetic and real datasets.

**Strengths:**

1. This work studies an important yet overlooked problem in programmatic weak supervision: the biases induced by unfair LFs
2. The proposed method, compatible with traditional fair ML methods, could mitigate biases and improve the performance at the same time.
3. The theoretical results are convincing, which show that the LF bias could be arbitrary yet can be fixed by the proposed method under some conditions

**Weaknesses:**

I am not aware of any major weakness except that the proposed model, if I understand it correctly, is a new label model built on an existing one, which means users have to use the proposed label model if they want to mitigate the biases. It is unclear how biases should be mitigated if users prefer other choice of label model.

In terms of label models that incorporate feature vector, one related work is missing: "Leveraging instance features for label aggregation in programmatic weak supervision"

**Questions:**

It is not intuitive how the improvement of fairness and performance can be achieved at the same time. Usually, there is a trade-off between performance and fairness in ML models, could you explain why and how which is not the case in this work?

**Limitations:**

see weakness above

---

> ### Author Rebuttal · Authors · 2023-08-08
>
> We appreciate the reviewer for the thoughtful comments and reference. We will add the suggested paper in our updated draft.
> * **On the choice of label models**: A remarkable property of our label model is that **it is only used for one step in our overall pipeline: mitigating source bias**. Once this step is complete, any other label model can be used for aggregating sources into pseudolabels. Our WRENCH experiment (Table 5) is conducted with the newly-introduced Hyper Label Model [1] after applying SBM.
> * **On the trade-off between performance and fairness in ML models**: We  discuss this question in our common response and provide more detail here.  The trade-off between performance and fairness in machine learning models is usually caused by inherent fairness violations in datasets. Models trained on such datasets are unfair, and so fairness techniques seek to mitigate this, but ultimately face constraints.
>
>     On the other hand, **our method tackles bias induced by creating labeled datasets via weak supervision**. Weak supervision allows for the efficient creation of labeled datasets by using multiple noisy sources of signal for labels. In addition to being noisy, these sources are often far more biased than hand-labeled datasets. As a result, a dataset created through weak supervision may prove to be highly unfair even if its counterpart with ground-truth labels was perfectly fair.
>
>     This problem is also an opportunity. Our approach is to model where bias is induced within weak supervision sources, remove this bias, and as a result, improve both fairness and performance, while keeping the advantage of using weak supervision---exploiting cheap label sources omnivorosly.
>
> [1] Wu, Renzhi, et al. "Learning Hyper Label Model for Programmatic Weak Supervision.", ICLR 2022.

---

> > ### Comment · Reviewer_6e6W · 2023-08-10
> > **Post rebuttal**
> >
> > I read the authors' rebuttal and other reviews. The author nicely addressed my questions. I would vote for accept and keep my score.

---

### Official Review · Reviewer_GLUc · 2023-06-30

**Soundness:** 3 good
**Presentation:** 3 good
**Contribution:** 3 good
**Rating:** 7
**Confidence:** 4

**Summary:**

*The paper studies the problem of unfairness introduced by weak supervision methods through noisy data augmentation. It proposes a mitigation strategy using counterfactuals that would create a more balanced/unbiased dataset.
* Paper shows that Labeling functions can be arbitrarily biased by preferring examples away from the center of the distribution and show theoretically there is no change in sample complexity required to achieve the gains in accuracy and fairness, under strict assumptions.
* Empirical results show that by further augmenting data that counterfactually transforms examples across groups, both accuracy and fairness metrics can be improved - with results on synthetic and popular benchmark fairness and weak supervision datasets.


**Strengths:**

* Strong empirical results on synthetic and real datasets
* New problem formulation that extends fairness methods to the weak supervision case

**Weaknesses:**

* Theoretical results are under strong distributional assumptions, how they can be relaxed should be better articulated
* Standard data augmentation techniques such as Autolabel [1] are missing in evaluation set up.

https://ieeexplore.ieee.org/abstract/document/10136178

**Questions:**

(see weaknesses)

**Limitations:**

This is missing, and should be included

---

> ### Author Rebuttal · Authors · 2023-08-08
>
> We thank the reviewer for the suggestions and noting the strengths of our work. We have added the suggested reference.
> * **On modeling assumptions**. Indeed, our label model _may appear_, at first glance, to require strong assumptions. However, in fact it **has weaker assumptions compared to previous weak supervision models.** We describe the differences versus two popular weak supervision models next. Common models use one of the following assumptions:
>
>     **i) Ising model assumption [1, 2, 3]**. Here, the model is given by
>     $$P(\lambda^1, \ldots, \lambda^m,y) = \frac{1}{Z}\exp(\theta_y y + \sum_{j=1}^m \theta_j \lambda^j y)$$
>     Here, the $\lambda$'s are labeling functions, $y$ is the latent true label, $Z$ is a partition function (a normalizing constant), the $\theta$'s are the model parameters.
>
>     This label model is based on the Ising pairwise interactions model. In this model, parameters are estimated globally, which means that **the label model requires uniform accuracy in the entire feature space---a strong assumption.**
>
>     In other words, regardless of the point $x \in \mathcal{X}$ whose label is being estimated, the assumption states that the labeling function accuracies are identical. Naturally, this is not a realistic assumption. Indeed, in practice, it has been observed that LF accuracy strongly depends on the input features, as certain parts of the feature space are invariably more challenging to label than others.
>
>
>     **ii) Strong smoothness leading to a partitionable feature space [4]**
>
>     $$P_x(\lambda^1, \ldots, \lambda^m, y) = \frac{1}{Z}\exp(\theta_y y + \sum_{j=1}^m \theta_{j,x} \lambda^j(x) y)$$
>     This model assumes that accuracy depends on the input feature $x$. In fact, in its most basic form, each $x$ has a separate model (with parameters given by $\theta_x$). However, since we can observe only one sample for each $x$, recovering parameters $\theta_{j, x}$ is impossible. Thus the authors in [4] instead assume strong smoothness: that is, **that the feature space can be partitioned and the accuracy in each partition is uniform.** This is still a strong assumption. While this model incorporated the input feature space, nevertheless accuracy tends to drop as data points get far from high accuracy centers---even within parts. This observation led to our newly-proposed model.
>
>     **iii) Proposed model**
>     $$P(\lambda^1(z), \ldots, \lambda^m(z), y) = \frac{1}{Z}\exp\left(\theta_yy + \sum_{j=1}^m \frac{\theta_{j}}{1+d(x^{\text{center}_j}, g_k(z))} \lambda^j(g_k(z)) y\right)$$
>     Our label model captures the phenomenon in Figure 2 of our draft, which shows that the accuracy of each LF drops as it moves away from its center. This means that our model neither require universal uniformity, as (i) does, nor per-part uniformity, as (ii) does. It makes the much weaker assumption of the existence of a center point where LFs are most accurate---which indeed matches practical LF scenarios, especially for LFs that are programs expressing heuristics. As a result, **our model not only has less restrictive assumptions, but also provides a framework that interprets unfairness as drift away from centers as an outcome of a group transformation.**
>
>
>
> [1] Ratner, Alexander, et al. "Snorkel: Rapid training data creation with weak supervision." VLDB 2017.
>
> [2] Fu, Daniel, et al. "Fast and three-rious: Speeding up weak supervision with triplet methods." ICML 2020.
>
> [3] Ratner, Alexander, et al. "Training complex models with multi-task weak supervision." AAAI 2019.
>
> [4] Chen, Mayee F., et al. "Shoring up the foundations: Fusing model embeddings and weak supervision.", UAI 2022.

---

### Official Review · Reviewer_UYjh · 2023-07-07

**Soundness:** 4 excellent
**Presentation:** 3 good
**Contribution:** 3 good
**Rating:** 7
**Confidence:** 4

**Summary:**

This paper proposes a novel bias mitigation technique to address the fairness issues in weak supervision settings. The core idea is to use a  counterfactual fairness-based correction method. The authors theoretically show that the proposed method can improve both accuracy and fairness and this is also supported by both synthetic and real datasets evaluations. Overall, this paper calls attention to bias and fairness studies in the weak supervision context, which has never been addressed specifically and provides an effective and theoretically sound method. To the best of my knowledge, the method considered in this paper is novel and is a valuable contribution to the community. Based on the above factors, I recommend acceptance.

**Strengths:**

This paper is well-motivated and well-written. The proposed method is intuitively simple yet very effective and theoretically sound. I appreciate the comprehensive evaluation both with synthetic data and with WRENCH.

**Weaknesses:**

This paper focuses on a novel area of weak supervision studies. Though the method proposed is simple, I do believe it has any major weak nesses.

**Questions:**

I welcome the authors to discuss the potential drawbacks of formulating the fairness argument behind counterfactual fairness, particularly in the programmatic weak supervision settings. Given diverse sources, wouldn't the biases be smoothed out?

**Limitations:**

Overall, this paper is well organized and I have not identify any apparent limitations for the current content.

---

> ### Author Rebuttal · Authors · 2023-08-08
>
> We are grateful to the reviewer for noting the strengths of our work and providing useful questions and comments.
> * **On the potential drawbacks of the proposed method in the programmatic WS settings**: A possible drawback of the proposed method is that it requires high quality estimation of Monge mapping (or any counterfactual mapping.) Fortunately, as we described in the general response, in the typical weak supervision settings, we have access to massive unlabeled datasets which enable reliable estimation in these procedures. Better yet, we have found that our method works well even with relatively small datasets ($n \leq 10^4$, e.g. the Wrench Mushroom dataset ($n=6499$)).
>
>     If the available unlabeled data is genuinely insufficient, our technique may struggle. To address this, we note that there are ways to overcome this limitation as well. For example, there are techniques that can dramatically reduce the amount of data that is needed in optimal transport. Such techniques involve the use of _keypoints_ (pairs of matched points) [1]. Domain experts could craft such pairs, enabling optimal transport, and ultimately our entire pipeline, to operate in the very low data regime.
>
> * **On the possibility that diverse label sources resolve unfairness naturally**: As mentioned in our common resonse, this appears to be a plausible approach towards mitigating unfairness. Unfortunately, diverse sources may not in fact behave in this way---further motivating the use of our approach. For example, our synthetic experiment in A.1. shows that diverse label sources can make fairness _worse_. In this setting, label sources that have high accuracy centers around the major clusters have higher weights in voting as an outcome of label model learning in weak supervision. Thus, increasing the number of LFs (diversifying) can be beneficial in finding such centers, and yields improved performance. However, it makes disadvantaged groups perform worse, since they are dislocated from high accuracy centers, leading to unfairness. SBM not only resolves such issues, but also further improves the performance gain by bringing back the dislocated data points around the high accuracy centers.
>
> [1] Gu, Xiang, et al. "Keypoint-guided optimal transport with applications in heterogeneous domain adaptation.", NeurIPS 2022.

---

### Official Review · Reviewer_wmSM · 2023-07-08

**Soundness:** 2 fair
**Presentation:** 2 fair
**Contribution:** 3 good
**Rating:** 5
**Confidence:** 4

**Summary:**

The paper focuses on the bias issue in weak supervision. The paper shows the labeling functions of weak supervision may produce biased pseudo-labels, which is first empirically shown in the paper. Also, they theoretically show that even though a dataset has fair ground-truth labels, the weak supervision process using labeling functions may produce biased weak labels for the dataset. To mitigate this issue, the paper proposes an optimal transport-based algorithm that modifies the weak labels to be fairer. In experiments, the paper evaluates the proposed algorithm in tabular, NLP, and computer vision datasets.

**Strengths:**

S1. The paper considers an important research problem, the fairness issue in the weak supervision pipeline. It seems this paper is the first work to handle fairness in weak supervision.

S2. The paper theoretically shows the labeling functions may produce unfair pseudo-labels even though the underlying label distribution is fair.

S3. The proposed method can be used together with the existing fair in-processing algorithms, which is another good aspect of the paper.


**Weaknesses:**

W1. The explanations on the labeling function itself are limited.
- For example, can the number of labeling functions affect the fairness performance? The Snorkel original paper [1] mentioned that the number of labeling functions highly affects the labeling performances, but the current paper does not explain how such details may make differences in the fairness scenario.
- Also, as shown in the vision dataset experiments, using fairer labeling functions can reduce the effectiveness of the proposed algorithm. Then, why is the proposed transport algorithm better than making labeling functions itself fairer? Currently, the paper seems to use very simple labeling functions (described in the appendix), and it would be helpful if the paper could provide any comparison between the proposed algorithm and another possible direction of making fairer labeling functions.

[1] Ratner et al., Snorkel: Rapid Training Data Creation with Weak Supervision, VLDB’18


W2. Experiments show some questionable results.
- As several scenarios show large F1 score drops, it is unclear whether it is okay to use the proposed algorithm. I understand the F1 score can be affected by the imbalanced label classes, but the F1 score degradation is still severe. For example, SBM (OT-S) + LIFT case in the Bank Marketing dataset and all SBM results in the Civil dataset show large F1 score drops. It would be helpful if the paper could provide at least some possible way to prevent such F1 score drops.
- In the experiments, no error range is provided, which makes the empirical results less convincing.


**Questions:**

Although the paper provides several meaningful discussions and insights, there are some remaining concerns, especially regarding the labeling function and empirical results. The details are in the above weakness section.

**Limitations:**

The paper did not discuss the limitations and potential negative social impact of the work. A possible limitation of this work can be other prominent fairness definitions that this work cannot handle.

---

> ### Author Rebuttal · Authors · 2023-08-08
>
> We sincerely thank the reviewer for their kind words, constructive feedback, and useful suggestions.
> * **On the number of labeling functions**: We discussed the relationship between the number of LFs, fairness, and performance in our common response. The result  shows that our method can resolve unfairness induced by weak supervision and improve performance, while existing weak supervision baselines lose fairness as they gain performance by increasing the number of LFs.
> * **On improving individual LF fairness**: Indeed, if we could write fair labeling functions directly, the resulting datasets would be fair as well. Unfortunately, this is extremely difficult. Most labeling functions are small programs written by domain experts to express heuristic ideas. Ensuring that these programs are themselves fair requires substantial research progress in other fields. Similarly, other sources for labeling functions are pre-existing external knowledge bases, crowdworkers, pretrained models, and more. Here as well, ensuring fairness directly would be very hard. As a concrete example, labeling functions in our vision dataset experiments are generated by models pretrained on other datasets. In such situations, where we do not have access to the original dataset, it may not even be possible to directly modify LFs to be fair. Instead, **our method provides a generic and cheap approach to deal with the bias of label sources in weak supervision.**
>
>     Another idea is to simply demand that each labeling function used is created to be fair in the first place, throwing away any type of source of LFs that may not satisfy this property (including heuristics, pretrained models, etc.) Doing so would ensure fairness as well---but removes all of the benefits of weak supervision. In fact, crafting only fair labeling functions that lead to a satisfactory dataset may be more difficult than hand-labeling in the first place, which is the pain point weak supervision was designed to solve. The **value of our approach is that it is best-of-all-worlds**: we can continue to omnivorously use weak sources of signal, while handling bias with the additional technique we proposed.
> * **On the F1 score in Bank Marketing and CivilComments datasets**: As mentioned, Bank Marketing and CivilComments datasets have severe class imbalance (the proportion of positive class: 11.8% and 11.3%, respectively), which may lead more imbalance in the transport step, producing high odds to flip noisy labels towards dominant classes. A simple remedy is performing balanced sampling. We can selectively use the target group data points so that their noisy labels have a balanced class ratio. The table below shows the result with balanced transport in Bank Marketing data with LIFT. We observe that this simple idea can improve the F1 score while maintaining accuracy and fairness scores. In general, _any_ counterfactual mapping estimation method that considers class imbalance can be plugged in.
>
> | Cond                                  | Acc           | F1            | DP Gap        | EO Gap        |
> | ------------------------------------- | ------------- | ------------- | ------------- | ------------- |
> | FS                                    | $0.912 \pm 0.000$ | $0.518 \pm 0.000$ | $0.128 \pm 0.000$ | $0.117 \pm 0.000$ |
> | WS (Baseline)                         | $0.674 \pm  0.000$ | $0.258 \pm 0.000$ | $0.543 \pm 0.000$ | $0.450 \pm 0.000$ |
> | SBM (w/o OT) + LIFT                   | $0.698\pm 0.000$ | $0.255 \pm 0.000$ | $0.088 \pm 0.000$ | $0.137 \pm 0.000$ |
> | SBM (OT-L) + LIFT                     | $0.892 \pm 0.015$ | $0.305 \pm 0.015$ | $0.104 \pm 0.001$ | $0.121 \pm 0.019$ |
> | SBM (OT-S) + LIFT                     | $0.698 \pm 0.011$ | $0.080 \pm 0.006$ | $0.109 \pm 0.017$ | $0.072 \pm 0.014$ |
> | SBM (OT-S) + LIFT + Balanced sampling | $0.827 \pm 0.002$ | $0.498 \pm 0.003$ | $0.133 \pm 0.002$ | $0.077 \pm 0.002$ |
>
> * **On error ranges**: Our proposed method is fairly simple and the effect of randomness is very limited. In repeated experiments with different seeds,  **we do not observe any significant deviations, but will provide error ranges from repeated runs in the updated draft.** We include several examples of these updated results in the following:
>
>     * Synthetic data experiments (Section 5.2. Figure 4): In rebuttal supplementary A.2, we show 95% confidence intervals, which are from 10 repetitions with different seeds. Unsurprisingly, the deviation rapidly drops as n increases.
>     * Real data experiments (Section 5.1.): We reported the mean of 5 repeated experiment results in Section 5.1. We will include error ranges in our updated manuscript. Examples include:
> #### Adult dataset
>
> | Cond                | Acc           | F1            | DP Gap        | EO Gap        |
> | ------------------- | ------------- | ------------- | ------------- | ------------- |
> | FS                  | $0.824 \pm 0.000$ | $0.564 \pm 0.000$ | $0.216 \pm 0.000$ | $0.331 \pm 0.000$ |
> | WS (Baseline)       | $0.717 \pm 0.000$ | $0.587 \pm 0.000$ | $0.475 \pm 0.000$ | $0.325 \pm 0.000$ |
> | SBM (w/o OT)        | $0.720 \pm 0.000$ | $0.592 \pm 0.000$ | $0.439 \pm 0.000$ | $0.273 \pm 0.000$ |
> | SBM (OT-L)          | $0.560 \pm 0.000$ | $0.472 \pm 0.000$ | $0.893 \pm 0.000$ | $0.980 \pm 0.000$ |
> | SBM (OT-S)          | $0.723 \pm 0.003$ | $0.590 \pm 0.003$ | $0.429 \pm 0.010$ | $0.261 \pm 0.005$ |
> | SBM (w/o OT) + LIFT | $0.704 \pm 0.000$ | $0.366 \pm 0.000$ | $0.032 \pm 0.000$ | $0.192 \pm 0.000$ |
> | SBM (OT-L) + LIFT   | $0.700 \pm 0.017$ | $0.520 \pm 0.005$ | $0.015 \pm 0.015$ | $0.138 \pm 0.020$ |
> | SBM (OT-S) + LIFT   | $0.782 \pm 0.043$ | $0.448 \pm 0.015$ | $0.000 \pm 0.000$ | $0.178 \pm 0.036$ |

---

### Author Rebuttal · Authors · 2023-08-08

We thank all of the reviewers for their kind comments and feedback. Reviewers recognized the strengths of our paper, which we briefly reiterate before we dive into in-depth responses.
* **This is the first study of fairness in weak supervision** (Reviewers wmSM, UYjh, GLUc, 6e6W).
* Our approach offers **theoretical results** showing that (1) even when a dataset with ground-truth labels is fair, **a weakly-supervised counterpart can be arbitrarily biased** and (2) a finite-sample **recovery result for the correction algorithm** (Reviewers wmSM, UYjh, GLUc, 6e6W).
* **Strong empirical results on synthetic and real datasets** (Reviewers UYjh, GLUc).
* The proposed method is **compatible with the existing fair machine learning algorithms** (Reviewers wmSM, 6e6W).

We address two common questions before proceeding to individual responses.
* **On advantages and limitations of our formulation** (Reviewers UYjh,  6e6W): First, we highlight how **bias in weak supervision differs** from typical supervised learning settings. In supervised learning, datasets have inherent fairness violations and fairness-aware methods try to train a fair model from unfair data. Such methods typically use constrained optimization, which tries to fit the training data under a fairness constraint. This usually entails tradeoffs between fairness and performance. The type of unfairness we tackle is **different**. It is an artifact of the WS process (i.e., how the noisy labels are generated) and may not be inherent in datasets (Theorem 1 in our work characterizes a range of such scenarios). As a result, it is possible to **i) maintain the advantages of using WS (cheap labeling), ii) improve fairness, and iii) improve performance**. Our method uses a simple counterfactual mapping to seek to achieve all of these benefits simultaneously.

    One potential drawback appears to be that high-quality estimation of the counterfactual mapping may be difficult. Fortunately, scenarios where this is the case are not common in weak supervision, since the standard setting involves using large unlabeled input datasets. This allows for small estimation error in the counterfactual mapping. As a result, we do not observe this limitation in any of the practical settings of interest. Even when this limitation is operative (i.e., when using small unlabeled datasets), it is still possible to overcome it by applying recent techniques in optimal transport [1,2] that enable improving the estimation of the counterfactual mapping.

    This is done, for example, by exploiting a handful of matched _keypoints_. Such methods can be easily applied if practitioners have some intuition about group matchings (e.g. toxic comment matching in different languages)---which is often the case in weak supervision scenarios.

* **On the relationship between the fairness and the number of labeling functions** (Reviewers UYjh, wmSM): Several reviewers hypothesize that diversifying label sources naturally resolves unfairness. Indeed, we had a similar belief early on in our study. Unfortunately, we ultimately observed the opposite scenario:  diversification can increase unfairness in practice. To demonstrate this idea, we conducted experiments varying the number of labeling functions and their parameters. We include one such case for synthetic data, depicting it in the figure in A.1. It shows that **in standard WS, increasing the number of LFs increases bias** as the accuracy improves. Meanwhile, **our approach improves fairness _and_ performance** when increasing the number of labeling functions.

[1] Gu, Xiang, et al. "Keypoint-guided optimal transport with applications in heterogeneous domain adaptation.", NeurIPS 2022.

[2] Panda, Nishant, et al. "Semi-supervised Learning of Pushforwards For Domain Translation & Adaptation." , arxiv 2023.

---

### Author Response · Authors · 2023-08-11
**Thank you again; further questions?**

Dear Reviewers,

We thank you again for your feedback, questions, and suggestions! We believe we have answered all of your questions in our responses and the updated draft. If you have additional questions, we would love to answer them!

The Authors

---

### Decision · Program_Chairs · 2023-09-21

**Decision:**

Accept (poster)

**Comment:**

All reviewers found the work interesting and relevant. After the author’s response, the reviewers did not see the need for further clarifications. All reviewers recommend acceptance.